# Glutamate neurotransmission from leptin receptor cells is required for typical puberty and reproductive function in female mice

**Cristina Sáenz de Miera[1], Nicole Bellefontaine[1], Susan J Allen[1], Martin G Myers[1,2,3], Carol F Elias[1,2,4]***

[1]Department of Molecular and Integrative Physiology, University of Michigan–Ann Arbor, Ann Arbor, United States; [2]Elizabeth W. Caswell Diabetes Institute, University of Michigan–Ann Arbor, Ann Arbor, United States; [3]Department of Internal Medicine, Division of Metabolism, Endocrinology and Diabetes, University of Michigan–Ann Arbor, Ann Arbor, United States; [4]Department of Obstetrics and Gynecology, University of Michigan–Ann Arbor, Ann Arbor, United States

*For correspondence:
cfelias@umich.edu

Competing interest: The authors declare that no competing interests exist.

**Abstract** The hypothalamic ventral premammillary nucleus (PMv) is a glutamatergic nucleus essential for the metabolic control of reproduction. However, conditional deletion of leptin receptor long form (LepRb) in vesicular glutamate transporter 2 (Vglut2) expressing neurons results in virtually no reproductive deficits. In this study, we determined the role of glutamatergic neurotransmission from leptin responsive PMv neurons on puberty and fertility. We first assessed if stimulation of PMv neurons induces luteinizing hormone (LH) release in fed adult females. We used the stimulatory form of designer receptor exclusively activated by designer drugs (DREADDs) in *Lepr*^Cre (LepRb-Cre) mice. We collected blood sequentially before and for 1 hr after intravenous clozapine-*N*-oxide injection. LH level increased in animals correctly targeted to the PMv, and LH level was correlated to the number of Fos immunoreactive neurons in the PMv. Next, females with deletion of *Slc17a6* (Vglut2) in LepRb neurons (*Lepr*^ΔVGlut2) showed delayed age of puberty, disrupted estrous cycles, increased gonadotropin-releasing hormone (GnRH) concentration in the axon terminals, and disrupted LH secretion, suggesting impaired GnRH release. To assess if glutamate is required for PMv actions in pubertal development, we generated a Cre-induced reexpression of endogenous LepRb (*Lepr*^loxTB) with concomitant deletion of *Slc17a6* (Vglut2^flox) mice. Rescue of *Lepr* and deletion of *Slc17a6* in the PMv was obtained by stereotaxic injection of an adeno-associated virus vector expressing Cre recombinase. Control *Lepr*^loxTB mice with PMv LepRb rescue showed vaginal opening, follicle maturation, and became pregnant, while *Lepr*^loxTB;Vglut2^flox mice showed no pubertal development. Our results indicate that glutamatergic neurotransmission from leptin sensitive neurons regulates the reproductive axis, and that leptin action on pubertal development via PMv neurons requires Vglut2.

## eLife assessment

This **important** study reports that glutamate signaling in LepRb PMv neurons is necessary for leptin-dependent fertility. The data supporting the conclusion is **solid**. This work will be of interest to researchers in the fields of both reproductive and metabolic biology.

## Introduction

Reproduction is strongly influenced by nutritional status due to the high energy demand required for this function. Food scarcity leads to suppression of fertility (*Ahima et al., 1996*; *Bronson and Marsteller, 1985*; *Cagampang et al., 1990*; *Cameron et al., 1991*; *Parfitt et al., 1991*). Similarly, excess adiposity may induce ovulatory dysfunction, higher risks of pregnancy complications and earlier age at puberty onset (*Biro et al., 2006*; *Burt Solorzano and McCartney, 2010*; *Mahany et al., 2018*). Metabolic cues signal energy status to the hypothalamo–pituitary–gonadal (HPG) axis, but the sites of action and mechanisms are not fully elucidated.

The adipocyte-derived hormone leptin is an essential metabolic signal acting in the brain to regulate the reproductive axis. Mice and humans with mutations in the leptin or leptin receptor genes are infertile and remain in a prepubertal state (*Bray and York, 1979*; *Farooqi et al., 2007*; *Farooqi et al., 2002*). Specific restoration of the leptin receptor (*Lepr*) gene only in neurons of *Lepr*-deficient mice rescues infertility (*de Luca et al., 2005*; *Quennell et al., 2009*). It is well accepted that leptin action in the modulation of the HPG axis is indirect. Gonadotropin-releasing hormone (GnRH) neurons do not express *Lepr* (*Quennell et al., 2009*) but they do receive direct innervation from *Lepr*-expressing neurons (*Louis et al., 2011*). The neural circuitry involved, however, is incompletely known.

An often-overlooked hypothalamic site, rich in neurons expressing the long form of LepR (LepRb), is the ventral premammillary nucleus (PMv) (*Donato et al., 2011*; *Han et al., 2020*; *Leshan et al., 2009*; *Louis et al., 2011*; *Ross et al., 2018*; *Scott et al., 2009*). In rats, bilateral lesions of the PMv disrupt estrous cycles and the ability of leptin to increase luteinizing hormone (LH) secretion following a fast (*Donato et al., 2009*). Endogenous restoration of LepRb exclusively in PMv neurons rescues pubertal maturation and fertility in *Lepr*-null female mice (*Donato et al., 2011*).

Leptin responsive neurons in the PMv project directly to and make contacts with GnRH cell bodies and their terminals in the adjacencies of the median eminence (*Boehm et al., 2005*; *Canteras et al., 1992*; *Donato et al., 2011*; *Leshan et al., 2009*). These cells also project to the anteroventral periventricular (AVPV) and arcuate (Arc) nuclei, home of kisspeptin (*Kiss1* gene) neurons, the main GnRH regulatory population (*Canteras et al., 1992*; *Donato et al., 2011*). Projections from the PMv make apparent contacts with *Kiss1* expressing cells in the AVPV (*Donato et al., 2011*), and stimulation of PMv neurons directly activates kisspeptin cells in both the AVPV and the Arc (*Ross et al., 2018*).

The PMv is essentially glutamatergic (*Vong et al., 2011*). LepRb neurons coexpress the vesicular glutamate transporter *Slc17a6* (Vglut2) (*Donato et al., 2011*), and acute administration of leptin depolarizes most of these neurons (*Williams et al., 2011*). Glutamatergic terminals are abundant in GnRH (*Moore et al., 2018*; *Yeo et al., 2021*) and in kisspeptin (*Porter et al., 2021*) neurons, while states of low leptin levels, as in fasting, reduce the number of glutamatergic synaptic inputs to kisspeptin neurons of male macaques (*Shamas et al., 2015*). Prior data, however, have shown that congenital deletion of *Lepr* in glutamatergic neurons has virtually no effect on pubertal development or reproductive function (*Martin et al., 2014*; *Zuure et al., 2013*).

In the attempt to untangle these apparent inconsistencies, we used chemogenetics to determine if remote stimulation of PMv neurons induces LH release in diestrous mice. We next used several transgenic mouse models to determine whether disruption of glutamate neurotransmission from *Lepr* neurons impairs leptin-induced puberty and normal reproductive function. Overall, our results indicate that glutamatergic signaling from leptin responsive PMv neurons is required for normal reproductive development and function.

## Results

### Chemogenetic activation of *Lepr^Cre* cells in the PMv induces LH secretion

To examine if activation of PMv LepRb neurons induces LH release in fed adult mice, we stereotaxically injected an adeno-associated virus vector (AAV) carrying the stimulatory form of designer receptor exclusively activated by designer drugs (DREADDs), hM3Dq, fused to mCherry (AAV-hM3Dq) unilaterally into the PMv of *Lepr^Cre* mice, followed by intravenous (iv.) injection of clozapine-*N*-oxide (CNO) or its metabolite, clozapine, after 4 weeks. Successful infection was confirmed by localized expression of mCherry and Fos immunoreactivity (ir) in the PMv following CNO or clozapine injection. For

**Table 1.** Experimental groups in the study of chemogenetic activation of *Lepr*^Cre cells in the ventral premammillary nucleus (PMv). Corresponding to **Figures 1–3**. The number of animals in each group is indicated in brackets.

| AAV vector | Drug | Injection site | LH increase 10–20 min | Group name | Color in Figure 1E–H, Figures 2 and 3 |
|---|---|---|---|---|---|
| | | | Yes (8) | 'PMv-hit' with LH increase |  |
| | | Hit in PMv (15) | No (7) | 'PMv-hit' with no LH increase |  |
| | | Missed PMv (2) | No | Negative controls |  |
| AAV-hM3Dq (18) | CNO | No Fos in mCherry cells. Removed from analysis (1). | | | |
| AAV-mCherry (5) | CNO | Hit in PMv | No | Negative controls |  |
| AAV-hM3Dq (4) | Clozapine | Hit in PMv | Yes | Clozapine hM3Dq |  |
| No AAV (7) | Clozapine | No injection | (2/7) | Clozapine no AAV |  |

clarification, **Table 1** shows the different experimental, positive and negative control groups, according to AAV and drug delivered, injection site and LH response.

From the AAV-hM3Dq animals that received CNO, 15 out of 18 *Lepr*^Cre diestrous females had successful AAV-hM3Dq injections (mCherry- and Fos-ir) in the PMv and were considered 'PMv-hits' (**Figure 1A, B**). Two mice showed injections outside the PMv (n = 2) and were considered 'PMv-misses'. The remaining mouse had a very small injection in the PMv (few mCherry-ir neurons, but no Fos-ir) and was removed from the analysis (**Table 1**). An additional group of five mice was injected with an AAV-mCherry vector and received iv. CNO, serving as negative controls. All five mice showed successful viral target in the PMv. For representation, this group is shown together with the PMv-misses as 'negative controls' (**Table 1**). An additional group of four mice carrying the AAV-hM3Dq received instead iv. clozapine to test the effect of the CNO metabolite. All four females showed successful viral target and Fos-ir in PMv neurons, and minimal Fos-ir in other hypothalamic sites (**Table 1**, **Figure 2A, B**). An additional negative control group (n = 7) did not receive an AAV injection and was injected with clozapine, to evaluate the effect of clozapine alone on LH release (**Table 1**).

About 55% (8/15) of mice with unilateral AAV-hM3Dq centered in the PMv showed an increase in LH release above 0.5 ng/ml within 10–20 min following the CNO injection (0.65–3.11 ng/ml). The remaining PMv-hit animals showed no increase in LH levels following the CNO injection. In PMv-miss animals, a modest increase in LH was observed in one animal (0.57 ng/ml) 10 min after the CNO injection and no increase was observed (under limit of detection) in the other. Only one of the 5 AAV-mCherry injected animals showed an increase in LH (0.93 ng/ml), but it was at 30 min following CNO (**Figure 1C, D**).

The area under the curve (AUC) was calculated for LH levels between 10 min before (−10) the injection until 20 min after the injection (time 0 was omitted since a few of the animals missed this sampling point) for all the groups, considering the point −10 min as baseline 0. The positive AUC for the PMv-hit with LH increase group was significantly different from 0, indicating a significant rise in LH levels during this time, while the positive AUC in the other groups did not differ from 0 indicating that LH did not increase following the injection (**Figure 1E, F**).

The number of Fos-ir neurons per section in the PMv was variable between groups, with higher number of Fos-ir neurons observed in PMv-hits that showed increased LH, than in animals that showed no LH increase and in negative controls (PMv-misses and AAV-mCherry combined) (**Figure 1G**). Peak LH levels (ng/ml) were positively correlated to the number of Fos-ir neurons in the PMv, suggesting a link between stimulation of LepRb neurons in the PMv and induction of LH release (**Figure 1H**). It also suggests that there is a threshold in which a number or a subset of PMv LepRb is necessary for successful stimulation of LH secretion.

All mice with AAV-hM3Dq targeting the PMv (Fos-ir) and injected with clozapine also showed an increase in LH 10 min after the injection (0.87–1.18 ng/ml, **Figure 2A, B**), whereas two out of seven (28.5 %) control mice with no AAV showed an increase in LH following the clozapine injection (**Figure 2C**). The positive AUC between −10 to 20 min showed a significant increase in LH levels in

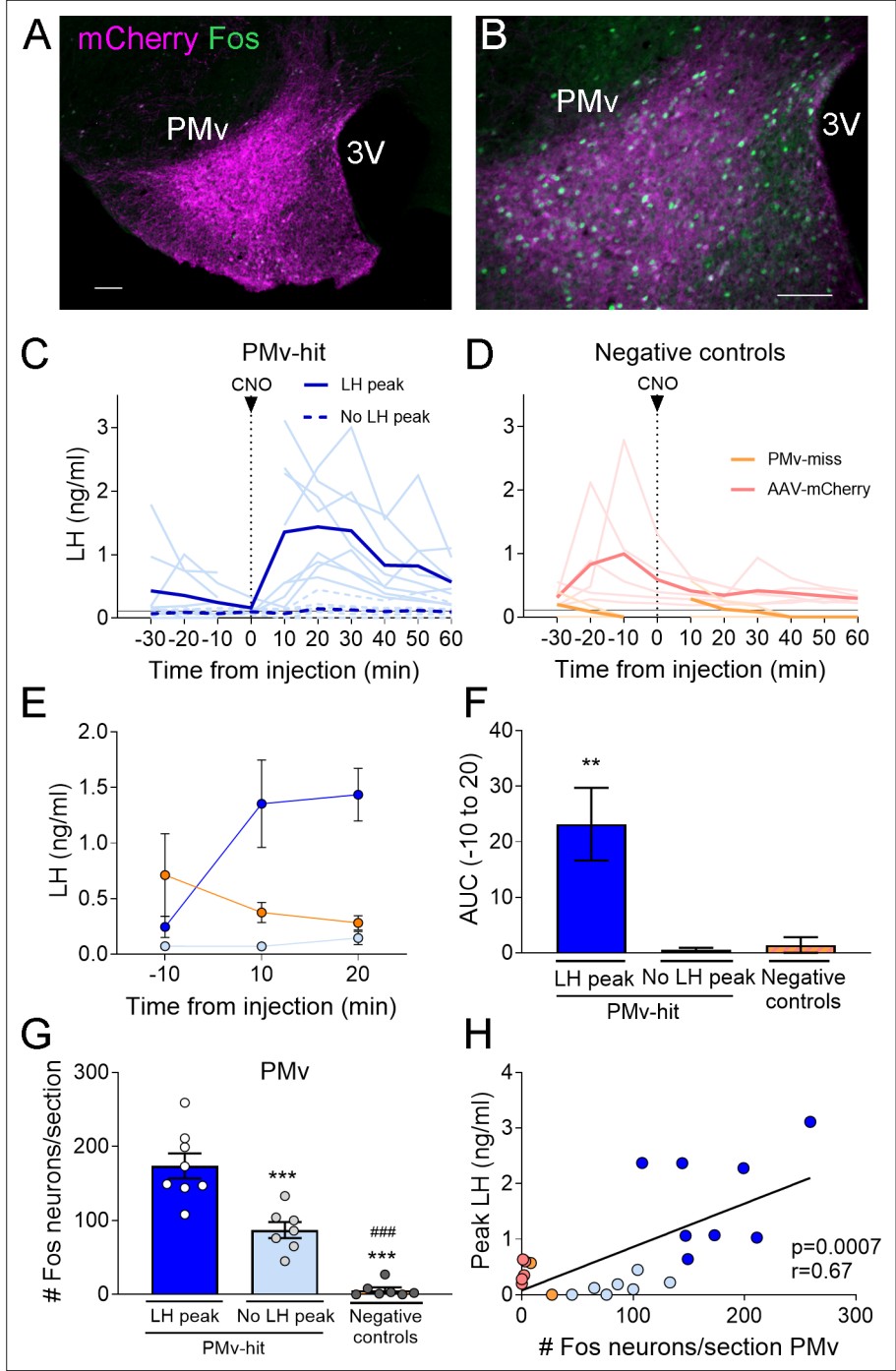

**Figure 1.** Chemogenetic activation of leptin receptor neurons in the ventral premammillary nucleus (PMv) induces luteinizing hormone (LH) release. (**A, B**) Representative fluorescent photomicrographs of unilateral AAV-hM3Dq PMv injection. Low (**A**) and high (**B**) magnification of an animal correctly targeted to the PMv. Magenta: mCherry-immunoreactivity (-ir); green: Fos-ir. (**C**) LH levels in 'PMv-hit' animals unilaterally expressing hM3Dq in the PMv following an intravenous (iv.) injection of clozapine-*N*-oxide (CNO) at time = 0 (*n* = 15). Continuous lines: animals showing an increase in LH had maximum LH levels ranging from 0.65 to 3.11 ng/ml. Dashed lines: animals showing no detectable increase in LH. Light blue lines represent individual values. Dark blue lines represent averaged values. (**D**) LH levels in negative control animals (*n* = 7), including 'PMv-miss' (orange) and AAV-mCherry (pink) animals following an iv. injection of CNO at time = 0. Light colored lines represent individual values. Dark colored lines represent average values. A gap is shown in the LH profile lines in animals that had a missing value at time 0 in C and D. (**E**) Average LH levels at −10, 10, and 20 min for all three groups: 'PMv-hit' with LH increase (*n* = 8, dark

*Figure 1 continued on next page*

*Figure 1 continued*

blue), 'PMv-hit' with no LH increase (*n* = 7, light blue), negative controls (including 'PMv-miss' and AAV-mCherry, *n* = 7, orange). (**F**) Positive area under the curve (AUC) of the values represented in E. One-sample *t*-tests ('PMv-hit' with LH increase: $t_7$ = 3.54, p = 0.009; 'PMv-hit' with no LH increase: $t_6$ = 1.37, p = 0.22; negative controls: $t_6$ = 1.00, p = 0.36), **p < 0.01 compared to '0'. (**G**) Number of Fos-ir neurons per section in the PMv in 'PMv-hit' animals with (*n* = 9) or without (*n* = 7) an increase in LH and in negative control animals, including 'PMv-miss' and AAV-mCherry animals (*n* = 7) 2 hr after the CNO injection (one-way analysis of variance [ANOVA], $F_{2,19}$ = 81.66; p < 0.0001). **p < 0.01 and ***p < 0.001 vs. PMv-hit LH increase group; ###p < 0.001 vs. PMv-hit no LH increase group. (**H**) Correlation between the number of Fos-ir neurons per section in the PMv and the peak LH level in the three groups (Pearson *r* = 0.67; p = 0.0007). 3V: third ventricle. Scale bars: 100 µm.

the clozapine AAV-hM3Dq group, but not in the group with no AAV (*Figure 2D, E*). The number of Fos-ir neurons per section in the PMv was higher in animals with hM3Dq than in those with no AAV, in which virtually no Fos-ir cells were observed (*Figure 2F*). Whether this variability is due to spontaneous LH release (*Czieselsky et al., 2016*) is not known. However, the consistently higher proportion of animals showing increased LH in AAV-hM3Dq PMv-hits injected with CNO (55 %) or clozapine (100 %) vs. PMv-misses and AAV-mCherry, and no-AAV control groups (0 and 28.5%, respectively), together with the strong activation of Fos in the PMv only in animals showing increased LH, make it unlikely that the observed rise in LH in experimental groups are a consequence of spontaneous LH pulsatile release.

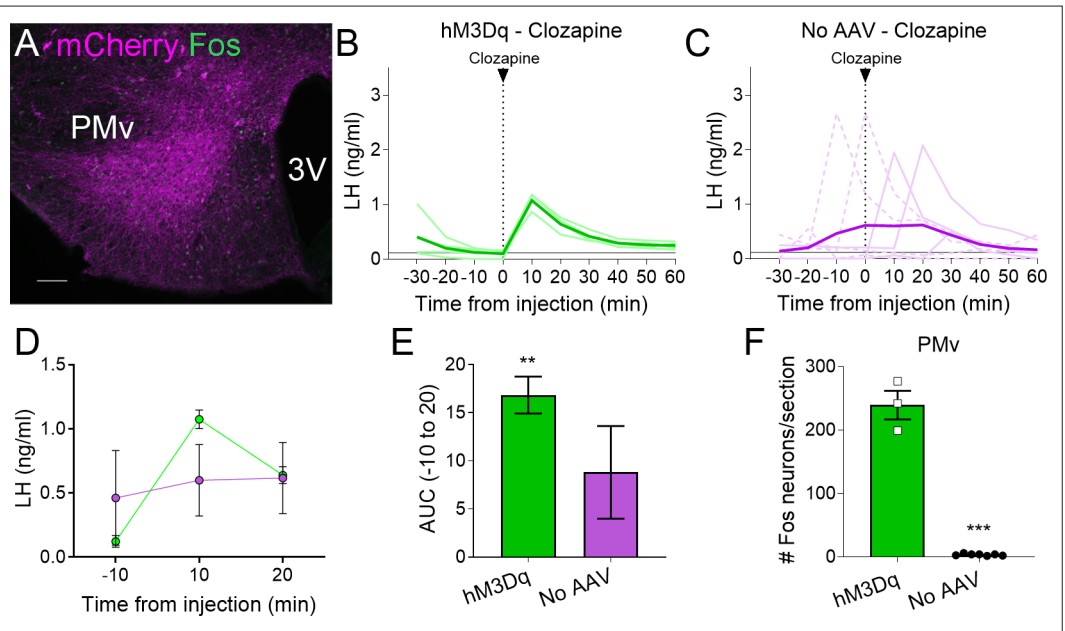

**Figure 2.** Clozapine induces Fos expression in the ventral premammillary nucleus (PMv) of animals expressing hM3Dq and luteinizing hormone (LH) release. (**A**) Representative fluorescent photomicrograph of a unilateral AAV-hM3Dq PMv injection in a clozapine injected animal. Magenta: mCherry-immunoreactivity (-ir); green: Fos-ir. (**B**) LH levels in animals unilaterally expressing hM3Dq in the PMv following an intravenous (iv.) injection of clozapine at time = 0 (*n* = 4). Light green lines represent individual values. Dark green line represents averaged values. (**C**) LH levels in animals not carrying any AAV following an iv. injection of clozapine at time = 0 (*n* = 7). Light magenta lines represent individual values. Dark magenta line represents averaged values. Continuous lines: animals with LH increase after the injection. Discontinuous lines: animals with no increase in LH after the injection. (**D**) Average LH levels at −10, 10, and 20 min for both groups injected with clozapine. hM3Dq: green; No AAV: magenta. (**E**) Positive area under the curve (AUC) of the values represented in D. One-sample *t*-tests (hM3Dq: $t_3$ = 8.79, p = 0.003; No AAV: $t_6$ = 1.83, p = 0.12), **p < 0.01 compared to '0'. (**F**) Number of Fos-ir neurons per section of the PMv in AAV-hM3Dq and no AAV animals 2 hr after a clozapine injection ($t_8$ = 17.46; p < 0.0001). ***p < 0.001. 3V: third ventricle. Scale bars: 100 µm.

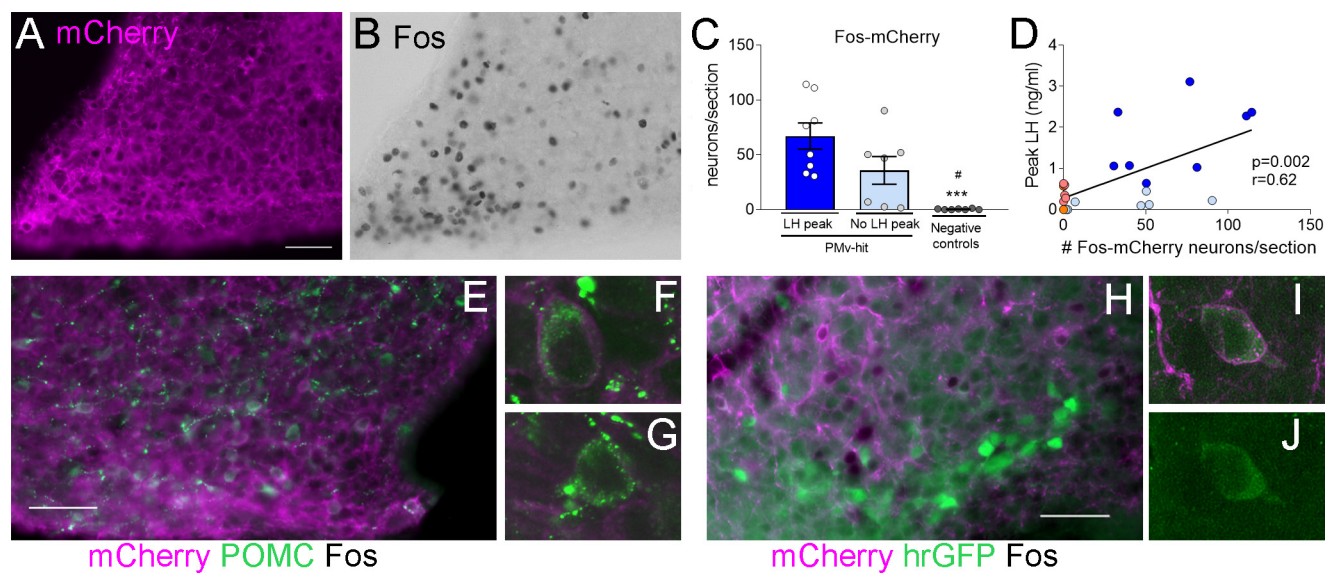

**Figure 3.** Clozapine-N-oxide (CNO) induction of luteinizing hormone (LH) secretion may have a mild contribution of pro-opiomelanocortin (POMC), but not KNDy, neurons. Representative images of (**A**) mCherry-immunoreactivity (-ir) and (**B**) Fos-ir of an animal with strong AAV contamination of the posterior Arc. (**C**) Number of Fos-ir neurons with mCherry-ir colocalization per section in the posterior Arc in the three groups 2 hr after the CNO injection. (Kruskal–Wallis test = 14.63; p < 0.0001), ***p < 0.001 vs. PMv-hit LH increase group, #p < 0.05 vs. PMv-hit no LH increase group. (**D**) Correlation between the number of mCherry-/Fos-ir neurons observed in the posterior Arc and the peak LH level in the three groups (Pearson r = 0.62; p = 0.002). (**E**) POMC-ir (green) in a mouse with viral contamination of the posterior Arc. Arrows: Fos-ir neurons (black) that coexpressed POMC and mCherry (magenta). Arrowheads: Fos-ir neurons that coexpressed POMC, but no mCherry, (**F**) Confocal high magnification image of a Fos-ir neuron coexpressing POMC and mCherry. (**G**) Confocal high magnification image of a Fos-ir neuron coexpressing POMC, but not mCherry. (**H**) GFP-ir (green) in a *Kiss1hrGFP* animal with viral contamination of the posterior Arc. Arrows: Fos-ir (black) neurons that coexpressed *Kiss1hrGFP* (green) and mCherry (magenta). (**I, J**) Confocal high magnification images of a Fos-ir neuron (arrow in H) coexpressing *Kiss1hrGFP* (green) and mCherry (magenta). Scale bars: 50 µm.

## AAV-hM3Dq contamination of the posterior Arc produces inconsistent CNO-stimulated LH secretion

As expected, the PMv-hit group showed some variability in the injection size, often contaminating to different extent LepRb neurons in areas adjacent to the PMv, including the posterior region of the Arc. Because this region has several neuronal populations involved in the control of reproduction, we evaluated the number of Fos-ir neurons in the Arc that coexpressed mCherry to assess if Arc neuronal activation was either direct or secondary to PMv neuronal activation (*Ross et al., 2018*). The total number of neurons coexpressing both Fos- and mCherry-ir per section in the posterior Arc was higher in PMv-hit animals, whether they showed an increase in LH, than in negative control animals (PMv-miss and AAV-mCherry combined) (*Figure 3A–D*). As observed for the PMv, peak LH levels were positively correlated to the number of Fos-ir + mCherry ir neurons in the Arc (*Figure 3D*), but not with the total number of Fos-ir neurons alone in the Arc (r = 0.3; p = 0.16). This observation suggested that LepRb-expressing neurons in the Arc may have contributed to the increase in LH observed following CNO treatment.

Within the Arc, pro-opiomelanocortin (POMC) neurons coexpress LepRb and have been shown to stimulate the reproductive axis (*Manfredi-Lozano et al., 2016*). To understand if direct activation of these neurons could have played a role in the induction of LH, we evaluated the percentage of activated neurons that expressed POMC peptide and the extent of mCherry colocalization in Fos-ir POMC neurons in PMv-hit group with higher contamination of the Arc, that also had some of the highest LH peaks, and in three hM3Dq clozapine animals with elevated LH levels but low mCherry contamination in the Arc. Coexpression of POMC-ir was observed in 9.3 ± 2.6% (12.3 ± 5.0 neurons) of the Fos-ir neurons of the posterior Arc in the PMv-hit animals and 9.6 ± 3.1% (2.3 ± 1.3 neurons) in the hM3Dq clozapine animals. Of these POMC-/Fos-ir neurons, about 50% also coexpressed mCherry-ir (7.3 ± 3.2 neurons) in the PMv-hit animals, but almost none in the hM3Dq clozapine animals (0.3 ± 0.3 neurons, *Figure 3E–G*). The small number of mCherry/Fos-ir neurons suggests that contamination

of POMC neurons may have only mild (if any) contribution to the higher increases in LH observed in some animals, but it is not determinant for inducing LH release.

We and others have previously shown that only about 10% of Arc Kiss1 (KNDy) neurons of female mice coexpress LepR (*Cravo et al., 2013*; *Donato et al., 2011*; *Louis et al., 2011*; *True et al., 2011*). To assess if contamination of KNDy neurons could have contributed to the induction of LH, we used *Lepr^Cre^; Kiss1^hrGFP^* mouse model to access the expression of Fos-ir and/or mCherry-ir in KNDy neurons following AAV-hM3Dq injections in the PMv (*n* = 4 PMv-hits, *Figure 1C*). In these animals, only 2.5 ± 0.95% of Fos-ir neurons in the posterior Arc were also *Kiss1^hrGFP^* positive (1.8 ± 0.3 neurons), and most of these expressed mCherry, suggesting that one or two KNDy neurons were directly activated by the virus. The low numbers of KNDy neurons expressing Fos-ir in *Kiss1^hrGFP^* PMv-hit mice indicates the increase in LH does not require KNDy neuronal activation (*Figure 3H–J*) in agreement with previous studies from our group showing that direct leptin action in Kiss1 neurons is neither required nor sufficient for puberty and reproduction (*Cravo et al., 2013*).

## Glutamate neurotransmission in LepRb cells is necessary for typical pubertal timing and estrous cycles

As most of LepRb PMv neurons are glutamatergic (*Donato et al., 2011*; *Zuure et al., 2013*), we generated mice that lack *Slc17a6* in LepRb neurons (*Lepr^ΔVglut2^* mice) to assess whether glutamatergic neurotransmission from LepRb cells is necessary for reproductive function.

As observed in a previous publication (*Tong et al., 2007*), virtually no *Slc17a6* expression was noticed in LepRb neurons of *Lepr^ΔVglut2^* mice (*Figure 4A–D*). Adult *Lepr^ΔVglut2^* mice (13–26 weeks old) developed late-onset obesity (*Figure 4E, F*) due to increased fat mass in both sexes (*Figure 4G, H*), reproducing the metabolic phenotype described before for these animals (*Xu et al., 2013*).

We then assessed if the lack of Vglut2 in LepRb cells alters pubertal maturation and adult reproduction. A thorough assessment of the metabolic and reproductive phenotype of the *Lepr^Cre^* mouse was performed before (*Garcia-Galiano et al., 2017*), and we found that they are indistinguishable from littermates negative for Cre. Female *Lepr^ΔVglut2^* and control *Lepr^Cre^* littermates had similar age of vaginal opening and showed no differences in body weight at puberty onset (*Figure 5A, B*). The day of first estrus, however, was delayed in *Lepr^ΔVglut2^* mice (*Figure 5C*). Of note, no differences in body weight were observed at vaginal opening (puberty onset) and first estrus (puberty completion) between experimental and control groups (*Figure 5D*). In males, balanopreputial separation (BPS) was monitored from P25 as a sign of advancement of sexual maturation. No differences in age at BPS were found between *Lepr^Cre^* and *Lepr^ΔVglut2^* littermates (*Figure 5E*). *Lepr^ΔVglut2^* male mice showed larger body weight than *Lepr^Cre^* mice at the time of BPS (*Figure 5F*).

Length of estrous cycles was longer (10–16 weeks of age, before the difference in body weight, *Figure 5G–I*) and the number of cycles completed in 30 days was lower in *Lepr^ΔVglut2^* mice compared to controls (*Lepr^Cre^*: 4.47 ± 0.30 cycles; *Lepr^ΔVglut2^*: 3.31 ± 0.36 cycles; $t_{26}$ = 2.45; p = 0.021). Female *Lepr^ΔVglut2^* spent similar time in estrus, less time in proestrus and more time in diestrus (*Figure 5J*). Analysis of female fertility showed that 75% of *Lepr^ΔVglut2^* against 100% of control mice produced at least one litter for 60 days since mating. Comparing fertile females, no differences were observed in latency to pregnancy (*Figure 5K*) or in the number of pups in the first litter (*Lepr^Cre^*: 7.86 ± 1.14 pups; *Lepr^ΔVglut2^*: 7.50 ± 1.41 pups; $t_{11}$ = 0.20; p = 0.85).

Because delay in first estrus, disruption of the estrous cycles and mild subfertility were observed before differences in body weight appeared (arrow in *Figure 4B*), our findings indicate that the disruption in reproductive function is caused by deficient glutamatergic neurotransmission in LepRb cells, not by metabolic dysregulation.

To gain insights into the underlying mechanisms associated with the disrupted reproductive function, we evaluated the HPG function. *Lepr^ΔVglut2^* females had denser GnRH-ir fibers in the Arc than Vglut2^flox^ control females (*Figure 6A–C*) suggesting retention of the neuropeptide at the neuronal terminals. To assess basal and stimulated GnRH release we measured basal LH and LH in response to intraperitoneal (ip.) injection of kisspeptin. Basal LH levels of diestrous females were not different between genotypes (*Figure 6D*). An immediate increase in LH (7 min) was observed following ip. kisspeptin in both groups, but LH levels in *Lepr^ΔVglut2^* mice were about half that observed in the controls (*Figure 6E*). Similarly, the overall LH AUC for the 30 minutes following the kisspeptin-10 injection in *Lepr^ΔVglut2^* females was about half the area observed in the controls (*Figure 6F*). These

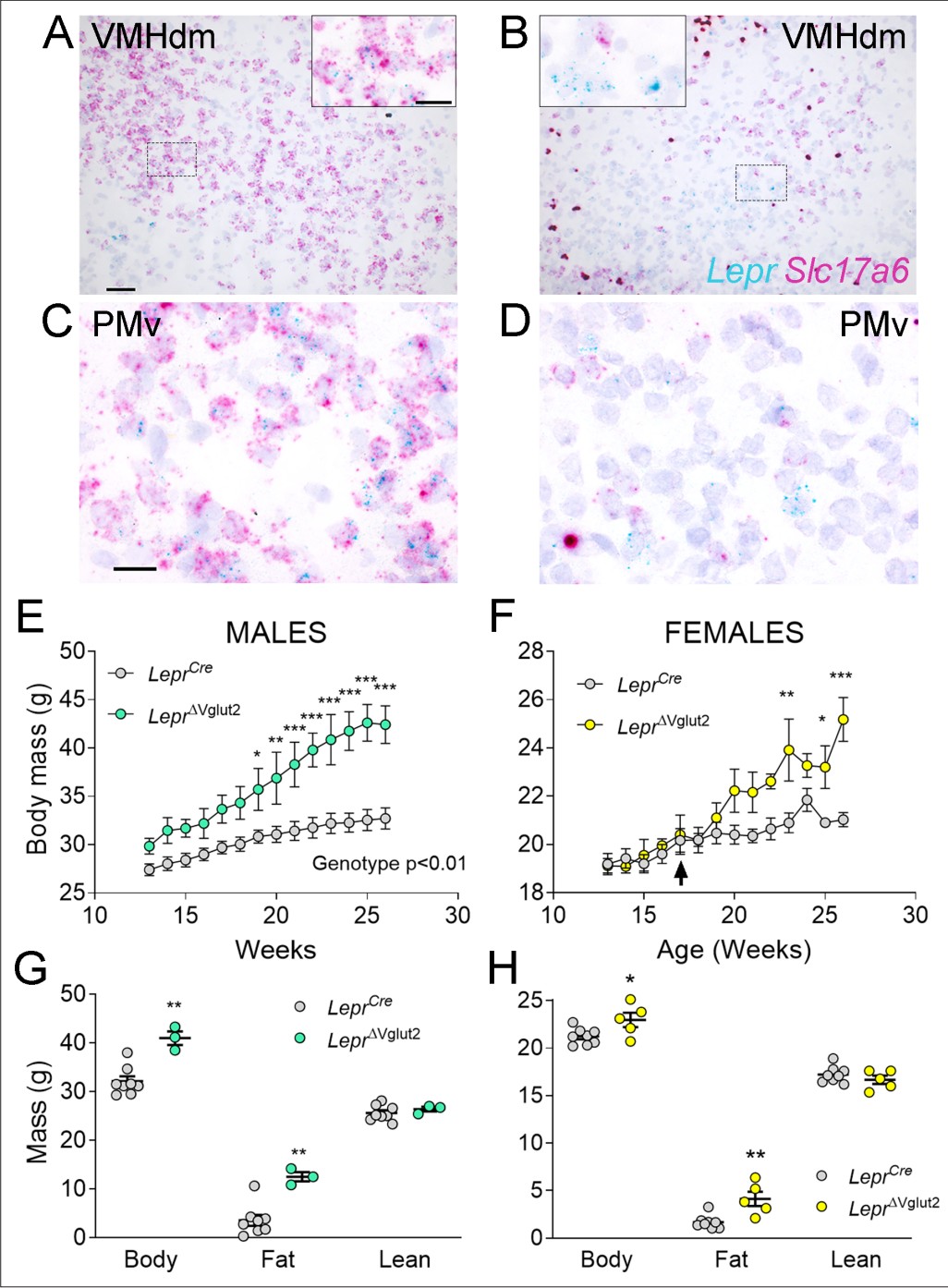

**Figure 4.** Adult $Lepr^{\Delta Vglut2}$ animals show an obese phenotype. (**A–D**) Micrographs of in situ hybridization showing colocalization between $Lepr$ (cyan) and $Slc17a6$ (magenta) gene expression. Dorsomedial region of the ventromedial hypothalamic nucleus (VMHdm) in control $Lepr^{Cre}$ (**A**), but no colocalization in experimental $Lepr^{\Delta Vglut2}$ (**B**) females. Insets show high magnification images for the area in the dashed rectangles (**C, D**). Ventral premammillary nucleus (PMv) at large magnification in control $Lepr^{Cre}$ (**C**), and experimental $Lepr^{\Delta Vglut2}$ (**D**). Scale bars: 50 μm for A, B and 20 μm for insets and C, D. (**E**) Body mass in adult $Lepr^{Cre}$ and $Lepr^{\Delta Vglut2}$ males (genotype main effect; $F_{1,10} = 13.1$; $p = 0.0047$, post hoc comparison: week 20: $t_{140} = 3.27$; $p = 0.019$; week 26: $t_{140} = 5.47$; $p < 0.0001$). (**F**) Body mass in adult $Lepr^{Cre}$ and $Lepr^{\Delta Vglut2}$ females (genotype main effect; $F_{1,11} = 4.34$; $p = 0.06$; post hoc comparisons: week 23: $t_{154} = 4.05$; $p = 0.0011$; week 25: $t_{154} = 3.1$; $p = 0.03$; week 26: $t_{154} = 5.59$; $p < 0.0001$). The arrow indicates the age at which estrus cycling follow-up finished in this cohort. (**G**) Body composition in males (fat mass: $t_9 = 3.77$; $p = 0.004$) and (**H**) in females (fat mass: $t_{11} = 3.7$; $p = 0.0035$). *$p < 0.05$, **$p < 0.01$, ***$p < 0.001$.

*Figure 4 continued on next page*

*Figure 4 continued*
The online version of this article includes the following source data for figure 4:
**Source data 1.** All data measured and analyzed for *Figure 4*.

results suggest a deficit in GnRH release despite higher accumulation of peptide at the terminals, which may be due to decreased pituitary LH content or disruption of hypothalamic regulation of GnRH release.

Pituitary expression of *Gnrhr*, *Lhb*, *Fshb* (*Figure 6G–I*) and the common glycoprotein alpha subunit (*Cga*, Vglut2$^{flox}$: 100 ± 17.47%; *Lepr*$^{\Delta Vglut2}$: 245.3 ± 89.91%; $t_8$ = 1.20; p = 0.26) relative to actin b (*Actb*) expression was similar between genotypes. Mediobasal hypothalamic expression of *Kiss1*, *Pdyn*, *Kiss1r*, and *Tac3r* relative to β-2-microglobulin (*B2m*) expression was similar between control and experimental females (*Figure 6J–M*). However, relative *Tac2* expression was about 50% lower in *Lepr*$^{\Delta Vglut2}$ compared to controls (*Figure 6N*).

## The action of LepRb PMv neurons in sexual maturation requires glutamate

In previous studies, we showed that endogenous reexpression of LepRb in the PMv-induced puberty and improved fertility of LepRb null mice (*Donato et al., 2011*; *Mahany et al., 2018*). To evaluate if glutamate neurotransmission is required for leptin-induced pubertal development, we produced a dual-floxed mouse model. *Lepr*$^{loxTB}$ mice that lack LepRb were crossed with a Vglut2$^{flox}$ line, to generate the *Lepr*$^{loxTB}$;Vglut2$^{flox}$ mice, in which stereotaxic delivery of AAV-Cre restores *Lepr* while deleting *Slc17a6* in the PMv.

The *Lepr*$^{loxTB}$ mice show no leptin-induced pSTAT3-ir (*Figure 7A, B, Q, S*). Expression of *Slc17a6* mRNA is intact in these mice despite the lack of leptin signaling (*Figure 7C, D*). These mice are obese and infertile, have reduced uterine size, and no ovarian corpora lutea (*Figure 7E–H, T*). As observed previously (*Mahany et al., 2018*), *Lepr*$^{loxTB}$ mice with reexpression of LepRb in the PMv (*Lepr*$^{loxTB}$ AAV-Cre) show leptin-induced pSTAT3-ir in the PMv (*Figure 7I, Q*). Expression of *Slc17a6* mRNA is intact in these mice (*Figure 7J*) and they have increased uterine size and ovarian corpora lutea (*Figure 7K, L, T*) indicating successful ovulation. The *Lepr*$^{loxTB}$ was used as the negative control, the *Lepr*$^{loxTB}$ with injection of AAV-Cre into the PMv was used as positive control for the procedure and the wild-type littermate was used as the positive control for the undisturbed morphology of the hypothalamus and the reproductive organs.

We obtained six *Lepr*$^{loxTB}$;Vglut2$^{flox}$ female mice with AAV-Cre centered in the PMv and two mice with AAV-Cre outside the PMv. Reexpression of LepRb was confirmed by the presence of leptin-induced pSTAT3-ir in the PMv neurons and the deletion of Vglut2 was confirmed by a decreased *Slc17a6* mRNA expression, compared to the *Lepr*$^{loxTB}$ with AAV-Cre. Mice with successful injections (*n* = 6) showed an increased number of leptin-induced pSTAT3-ir neurons, similar to that obtained in AAV-Cre injected *Lepr*$^{loxTB}$ (*Figure 7M, Q*). Mice with AAV-Cre injections outside the PMv (*n* = 2) showed pSTAT3-ir in adjacent nuclei, that is, dorsomedial and ventromedial nuclei of the hypothalamus and were removed from the analysis. Variable contamination of the Arc was evident but the number of pSTAT3-ir was similarly negligible in AAV-Cre injected *Lepr*$^{loxTB}$ and *Lepr*$^{loxTB}$;Vglut2$^{flox}$ mice (*Figure 7I, M, S*). Comparing AAV-Cre injected groups, *Slc17a6* mRNA was decreased only in *Lepr*$^{loxTB}$;Vglut2$^{flox}$ mice with injections successfully targeting PMv neurons (*Figure 7N, R*).

Correctly PMv injected *Lepr*$^{loxTB}$;Vglut2$^{flox}$ mice showed no sign of puberty onset (vaginal opening) and no pregnancy until the end of the experiment (about 60 days). Histological examination showed these mice had reduced uterine size and no corpora lutea suggesting low estradiol, lack of sexual maturation and ovulation (*Figure 7O, P, T*). Body weight in LepR null mice with reactivation of LepR in PMv neurons was not impacted as published before (*Donato et al., 2011*; *Mahany et al., 2018*). Similarly, the progression of body weight in mice with reactivation of LepRb and deletion of Vglut2 in PMv neurons was not affected by this deletion.

These findings demonstrate that glutamate neurotransmission is required for leptin action in pubertal development and reproductive function mediated by PMv neurons.

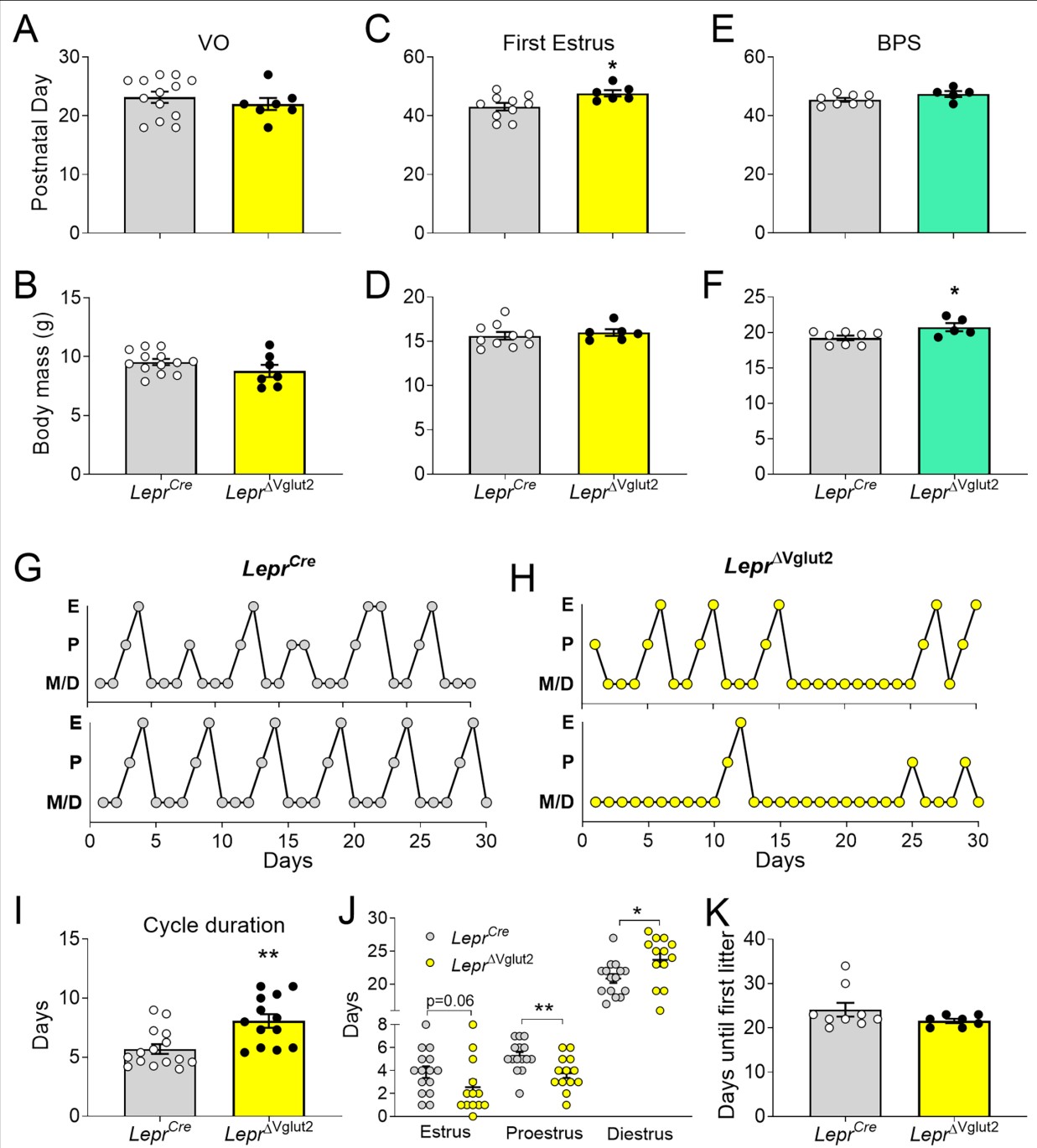

**Figure 5.** Reproductive development and estrus cycles are altered in $Lepr^{\Delta Vglut2}$ females. (**A**) Age of vaginal opening (VO; $t_{18} = 0.84$; $p = 0.41$) and (**B**) body mass at vaginal opening (VO) ($t_{18} = 1.47$; $p = 0.16$) in females. (**C**) Age of first estrus ($t_{14} = 2.45$; $p = 0.028$) and (**D**) body mass at first estrus ($t_{14} = 0.59$; $p = 0.56$) in females. (**E**) Age of complete balanopreputial separation (BPS; $t_{11} = 1.80$; $p = 0.1$) and (**F**) body mass at BPS ($t_{11} = 2.43$; $p = 0.033$) in males. (**G**) Representative estrus cycle profiles from control $Lepr^{Cre}$ and (**H**) experimental $Lepr^{\Delta Vglut2}$ females (E: estrus, P: proestrus, M/D: metestrus/diestrus). (**I**) Cycle duration in adult females ($t_{25} = 3.71$; $p = 0.001$). (**J**) Time spent in each phase of the cycle during the 30 days studied. Estrus ($t_{26} = 1.97$; $p = 0.059$), proestrus ($t_{26} = 2.81$; $p = 0.009$), and diestrus ($t_{26} = 2.39$; $p = 0.024$). (**K**) Days spent from mating until the first litter was observed in females paired with proven male breeders (Mann–Whitney test; $p = 0.32$). *$p < 0.05$, **$p < 0.01$.

The online version of this article includes the following source data for figure 5:

**Source data 1.** All data measured and analyzed for *Figure 5*.

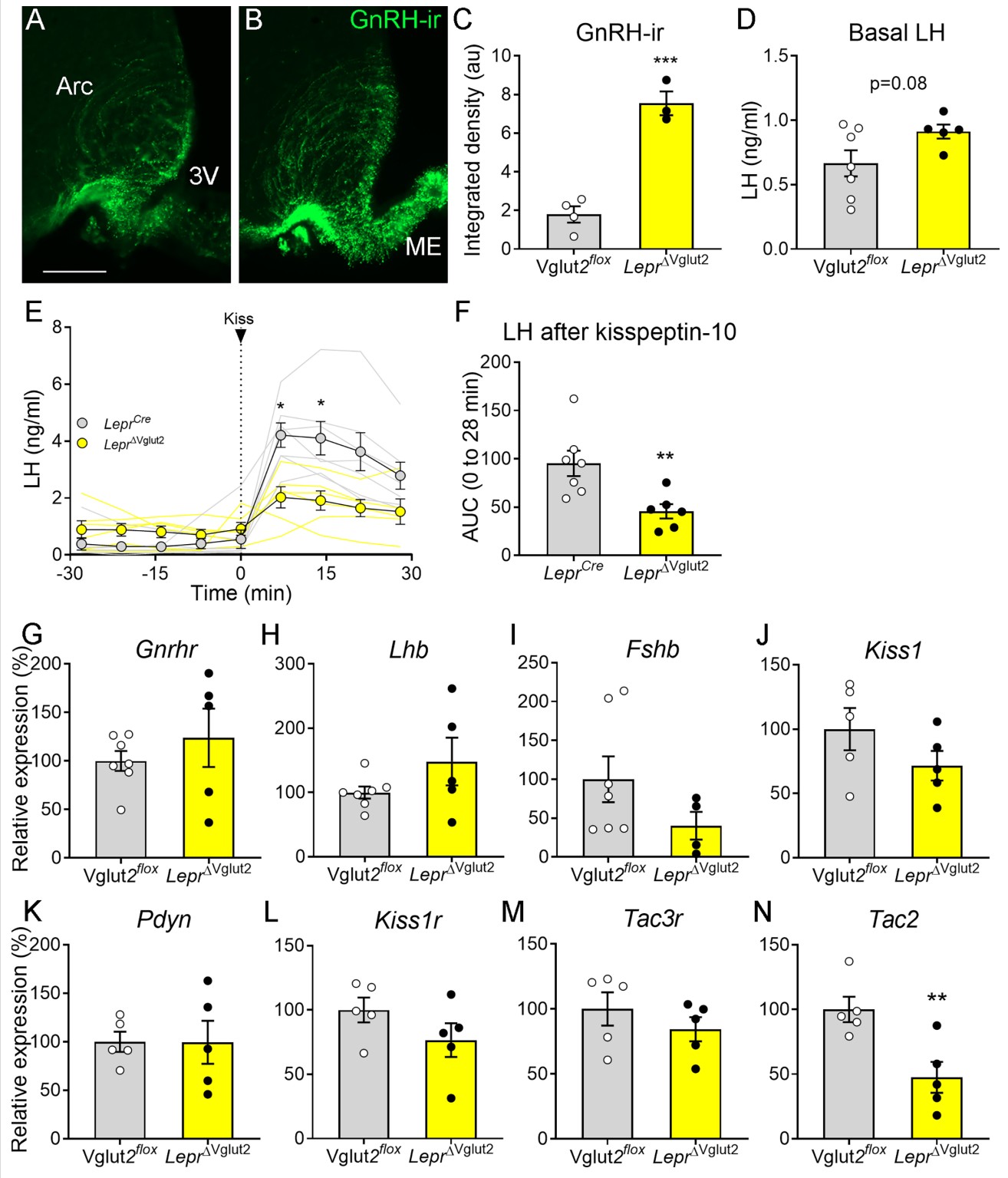

**Figure 6.** Gonadotropin-releasing hormone (GnRH) content in terminals and luteinizing hormone (LH) secretion are altered in *Lepr*^ΔVglut2 female mice. Representative fluorescent photomicrograph of GnRH-immunoreactivity (-ir) in the arcuate nucleus (Arc) and median eminence (ME) in (**A**) control Vglut2^*flox* and (**B**) *Lepr*^ΔVglut2 diestrous females. Scale bar: 100 μm. (**C**) GnRH-ir integrated density in the Arc in Vglut2^*flox* (*n* = 4) and *Lepr*^ΔVglut2 (*n* = 3) females ($t_5$ = 8.01; p = 0.0005). (**D**) Basal LH levels in Vglut2^*flox* (*n* = 7) and *Lepr*^ΔVglut2 (*n* = 5) females ($t_{10}$ = 1.91; p = 0.08). (**E**) LH levels in *Lepr*^*Cre* (*n* = 7) and *Lepr*^ΔVglut2 (*n* = 6) females before and after a kisspeptin-10 (65 μ/kg) intraperitoneal (ip.) injection of kisspeptin at time = 0. Light gray: individual *Lepr*^*Cre* females, Light yellow: individual *Lepr*^ΔVglut2 females. After the injection (time 7–28), LH levels were higher in control than in floxed females Mixed effects

*Figure 6 continued on next page*

*Figure 6 continued*

model: main effect for 'time' $F_{1.14, 12.19}$ = 15.91, p = 0.001; 'genotype' $F_{1, 11}$ = 8.26, p = 0.015; 'time × genotype' interaction: $F_{3, 32}$ = 5.00, p = 0.006; Sidak's post hoc effect for 7 min ($t_{10.97}$ = 3.79, p = 0.01) and for 14 min ($t_{9.4}$ = 3.22, p = 0.04). (**F**) Area under the curve (AUC) of LH levels after the kisspeptin injection ($t_{11}$ = 3.14; p = 0.009). (**G**) *Gnrhr* ($t_{10}$ = 0.86, p = 0.41), (**H**) *Lhb* ($t_{10}$ = 1.47, p = 0.17), and (**I**) *Fshb* ($t_9$ = 1.63, p = 0.14) relative to Actin B (*Actb*) mRNA levels in the pituitary of Vglut2$^{flox}$ (*n* = 7) and Lepr$^{ΔVglut2}$ (*n* = 5) females. (**J**) *Kiss1* ($t_8$ = 1.42, p = 0.19), (**K**) *Pdyn* ($t_8$ = 0.02, p = 0.98), (**L**) *Kiss1r* ($t_8$ = 1.44, p = 0.19), (**M**) *Tac3r* ($t_8$ = 0.99, p = 0.35) and (**N**) *Tac2* ($t_8$ = 3.39, p = 0.009), relative to β-2-microglobulin (*B2m*) mRNA levels in the mediobasal hypothalamus of Vglut2$^{flox}$ (*n* = 5) and Lepr$^{ΔVglut2}$ (*n* = 5) females. *p < 0.05; **p < 0.01; ***p < 0.001.

The online version of this article includes the following source data for figure 6:

**Source data 1.** All data measured and analyzed for *Figure 6*.

## Discussion

Understanding the role of leptin in the control of reproduction has largely advanced in the last few years, but many open questions remain regarding the neural pathways and the molecular mechanisms involved. While overlapping neural pathways exist, the PMv has arisen as an essential hub in this circuitry. The PMv is a glutamatergic nucleus (*Donato et al., 2011*; *Vong et al., 2011*) rich in leptin-responsive cells (*Leshan et al., 2009*; *Louis et al., 2011*; *Scott et al., 2009*; *Williams et al., 2011*). Nevertheless, recent studies have shown that deletion of LepRb in glutamatergic cells does not disrupt puberty or fertility, as opposed to large reproductive deficits when LepRb is deleted from GABAergic cells (*Martin et al., 2014*; *Zuure et al., 2013*). We have therefore investigated in more detail the role of glutamatergic neurotransmission in LepRb PMv neurons.

Leptin treatment prevents a fall in LH levels in states of negative energy balance in rodents and humans (*Ahima et al., 1996*; *Chan et al., 2003*; *Donato et al., 2009*; *Welt et al., 2004*). In ob/ob (Lep$^{ob}$) mice, which are infertile and hypogonadotropic, leptin administration increases LH levels, while this treatment is ineffective in PMv lesioned Lep$^{ob}$ mice (*Donato et al., 2011*), indicating that intact PMv is required for the leptin stimulatory effect on LH in leptin-deficient mice. In the current study, we expanded this observation showing that in fed mice, acute unilateral activation of leptin responsive PMv neurons using stimulatory DREADDs transiently elevated LH levels in a normal nutritional state (i.e., not in negative energy balance). This increase was correlated with the number of activated PMv LepRb neurons, and it was only observed following a substantial neuronal stimulation (Fos activation). The animals that did not show an LH increase despite having the virus targeted to the PMv had low Fos-ir in the PMv, possibly due to virus inactivity or low CNO entry into the brain (*Gomez et al., 2017*).

It is well accepted that clozapine has high affinity to DREADDs (*Gomez et al., 2017*). Thus, the hM3Dq injected animals showing clozapine-induced LH release strongly support our observations using CNO, that is targeted activation of PMv neurons leads to increase in circulating LH. Of note, optogenetic activation of a pituitary adenylate-cyclase-activating polypeptide (PACAP) expressing subpopulation of PMv neurons leads to increased activity of kisspeptin neurons. Whether this increases LH release is not known. Instead, the authors showed that deletion of PACAP in the PMv leads to an exacerbation of LH response to acute leptin administration (*Ross et al., 2018*), what reinforces the role of LepRb PMv neurons in LH secretion. Further studies are required to define the interaction between glutamate and PACAP neurotransmission in downstream targets of PMv neurons.

The use of stereotaxic injections inevitably leads to heterogeneity in the injection sites. The main caveat to our study was the activation in some animals of neighboring hypothalamic areas containing leptin receptor, abundant in the mediobasal hypothalamus (*Louis et al., 2011*; *Scott et al., 2009*). Larger injections in a few animals, and perhaps the presence of dendrites from LepRb neurons adjacent to PMv, led to contamination of other sites, mainly the Arc. For this reason, we investigated the distribution of Fos-ir in Arc neuronal populations involved in stimulation of LH secretion (i.e., POMC and kisspeptin).

POMC neurons have been involved in modulation of the reproductive axis (*Leranth et al., 1988*; *Manfredi-Lozano et al., 2016*; *Scimonelli and Celis, 1990*). The POMC-derived peptide α-melanocyte-stimulating hormone has a stimulatory role on reproduction via actions on kisspeptin and GnRH neurons (*Manfredi-Lozano et al., 2016*; *Roa and Herbison, 2012*). Therefore, direct activation of POMC neurons, although small in number, may have contributed to the higher LH peak observed in these animals. On the other hand, kisspeptin is a direct stimulator of GnRH but only about 10% of KNDy neurons in female mice express LepRb (*Cravo et al., 2013*; *Cravo et al., 2011*;

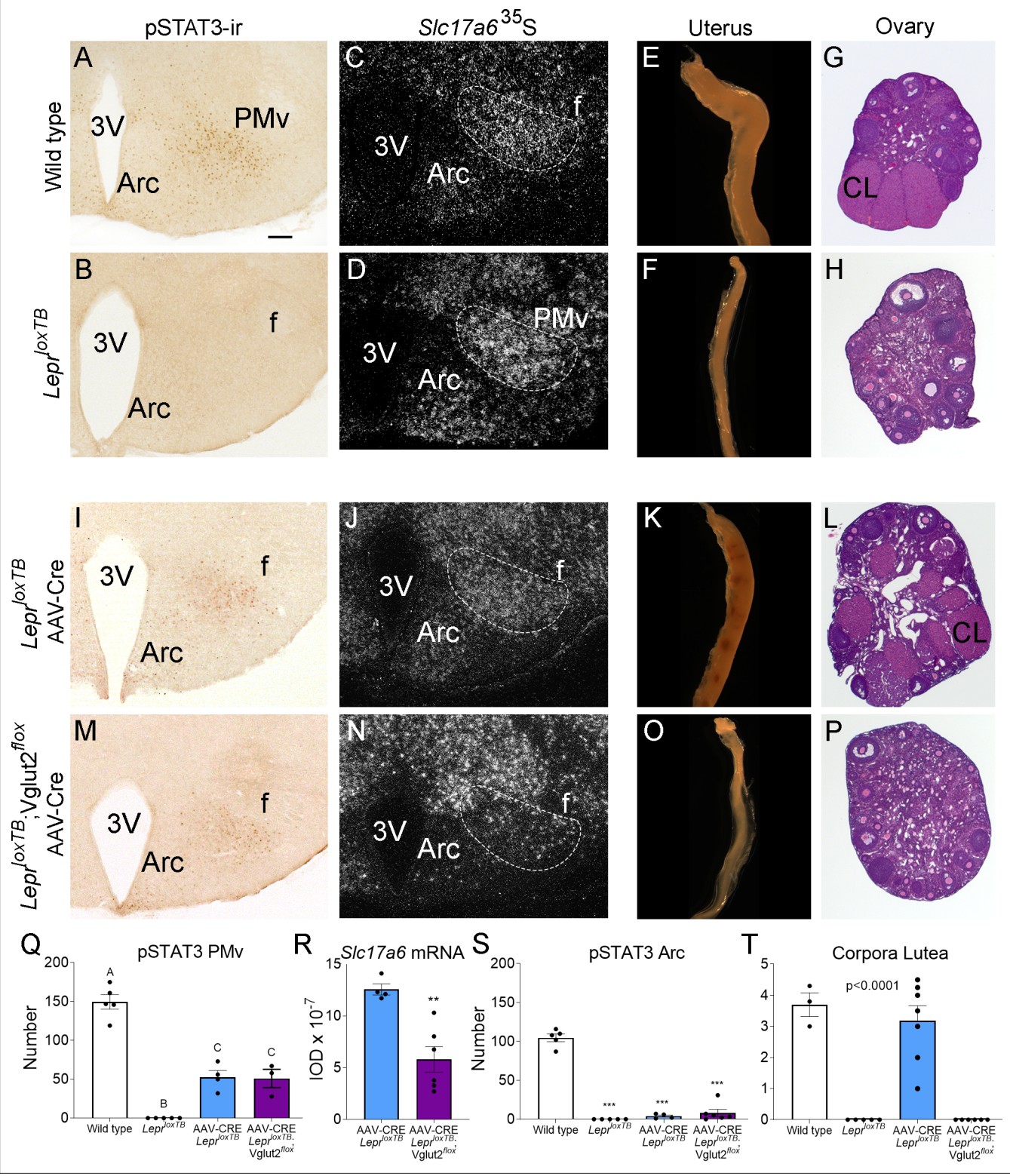

**Figure 7.** Glutamate signaling from the ventral premammillary nucleus (PMv) is required for leptin effect on pubertal development in female mouse. (**A, B, I, M**) Microphotographs of the PMv showing pSTAT3-ir following leptin administration. Note lack of pSTAT3-ir in *Lepr^loxTB^* mouse. (**C, D, J, N**) Darkfield microphotographs showing *Slc17a6* mRNA (silver grains) in the PMv. Note decreased *Slc17a6* mRNA expression in *Lepr^loxTB^*;Vglut2^flox^ mice injected with AAV-Cre. (**E, F, K, O**) Microphotographs of representative single uterine horns. Note lack of uterine growth in *Lepr^loxTB^* mouse and in *Lepr^loxTB^*;Vglut2^flox^ mouse injected with AAV-Cre. (**G, H, L, P**) Microphotographs of representative ovary sections. Note lack of *corpora lutea* in *Lepr^loxTB^*

*Figure 7 continued on next page*

*Figure 7 continued*

mouse and in $Lepr^{loxTB}$;Vglut2$^{flox}$ mouse injected with AAV-Cre. 3V: third ventricle, Arc: arcuate nucleus, f: fornix, CL: *Corpus luteum*. Scale bars: 100 μm. (**Q**) Number of pSTAT3 expressing cells in the PMv in wild-type (*n* = 5), $Lepr^{loxTB}$ (*n* = 5), AAV-CRE injected $LepR^{loxTB}$ (*n* = 4), and AAV-CRE injected $Lepr^{loxTB}$;Vglut2$^{flox}$ (*n* = 3) ($F_{3,13}$ = 73.53, p < 0.0001). Different capital letters indicate groups that are significantly different to the others (at least p<0.01) (**R**) Quantification of the *Slc17a6* hybridization signal in the PMv AAV-CRE injected $Lepr^{loxTB}$ (*n* = 4), and AAV-CRE injected $Lepr^{loxTB}$;Vglut2$^{flox}$ (*n* = 6) ($t_8$ = 4.20; p = 0.003). (**S**) Number of pSTAT3 expressing cells in the Arc ($F_{3,16}$ = 171.8, p < 0.0001). (**T**) Number of corpora lutea in the ovary ($F_{3,17}$ = 31.16, p < 0.0001). **p < 0.01; ***p < 0.001.

The online version of this article includes the following source data for figure 7:

**Source data 1.** All data measured and analyzed for *Figure 7*.

*Donato et al., 2011*; *Louis et al., 2011*; *True et al., 2011*). Restoration of LepRb in kisspeptin neurons does not restore fertility in *Lepr*-deficient mice (*Cravo et al., 2013*; *Cravo et al., 2011*). The very low number of Fos-ir KNDy neurons we observed makes it unlikely that contamination or downstream activation of KNDy neurons contributed to the LH secretion. Of note, in kisspeptin signaling deficient mice, glutamate receptor agonists induce LH secretion, indicating that alternative glutamatergic pathways bypass KNDy neurons to regulate the reproductive axis (*García-Galiano et al., 2012*). Alternatively, chemogenetic activation of leptin sensitive PMv neurons might primarily act via direct activation of GnRH cell bodies in the preoptic region or their terminals in the median eminence, as discussed below for the $Lepr^{\Delta Vglut2}$ females. Uncharacterized mCherry-expressing Fos neurons were observed in the posterior Arc, but lack of prior knowledge of a role in reproduction curtails the evaluation of a plausible role in LH release.

Due to lack of a specific model to target the PMv neurons, we investigated the reproductive phenotype in animals with Vglut2 deletion in LepRb expressing neurons, rendering them incapable of transmitting fast-acting downstream signals (*Xu et al., 2013*). These animals develop obesity (*Xu et al., 2013*). To avoid the effects of high adiposity in reproductive outcome, body weight was monitored during all reproductive phenotype assessment. While the reproductive effect in males was not obvious, $Lepr^{\Delta Vglut2}$ females showed delayed pubertal completion and blunted estrous cycles. The reproductive deficits were observed prior to the manifestation of any changes in body weight, indicating that these alterations are independent of metabolic dysregulation (*Donato et al., 2011*). It is important to stress that Vglut2 is expressed in several other LepRb neurons including the dorsomedial and the ventromedial nuclei of the hypothalamus (*Xu et al., 2013*). Based on extensive literature and circuit tracing, the PMv remains the prime candidate to relay leptin's effect in reproductive function.

To test if glutamatergic neurotransmission in LepRb PMv neurons is required for sexual maturation we used LepR-null;Vglut2-floxed females. This strategy aimed to 'bypass' the potential redundancy played by alternative neural pathways. Our prior studies showed that selective reexpression of LepRb in the PMv-induced puberty and fertility in LepR-null female mice (*Donato et al., 2011*; *Mahany et al., 2018*). Here, we observed that deletion of Vglut2 in PMv neurons blocked this effect. Female gonads remained in a prepubertal state, with undeveloped uterine horns and no ovarian corpora lutea.

Despite prior reports showing that lack of leptin signaling in glutamatergic cells has no effects on reproduction (*Martin et al., 2014*; *Zuure et al., 2013*), we show that glutamatergic neurotransmission in leptin responsive cells of the PMv is necessary for typical pubertal development and reproduction. Due to the fundamental role of reproduction for species survival, redundancy of neural pathways is expected, and developmental adaptations may have occurred in those studies. Of note, a recent study showed that reexpression of LepRb only in GABAergic cells is not sufficient for restoration of reproduction in otherwise LepR-null mice, emphasizing the need for other populations (*Quaresma et al., 2021*). Similarly, alternative metabolic signals may take over control in the absence of leptin signaling in this specific pathway (*Hill et al., 2010*; *Hill and Elias, 2018*).

Leptin depolarizes about 75% of PMv LepRb neurons (*Williams et al., 2011*). Depolarizing glutamatergic neurons may directly activate downstream neurons. The LepRb PMv neurons project to the preoptic area, where they directly contact GnRH neurons (*Leshan et al., 2009*; *Louis et al., 2011*). Alternatively, LepRb neurons may activate astrocytes, which in turn stimulate GnRH neurons and LH pulsatility (*Vanacker et al., 2021*) or GnRH fibers at the median eminence (*Donato et al., 2011*). In fact, $Lepr^{\Delta Vglut2}$ females showed increased GnRH-ir fibers located in the adjacencies of the median eminence. In these females, LH release in response to kisspeptin was reduced suggesting peptide retention and deficits in GnRH release. Similar results were obtained after reactivation of leptin signaling in the PMv

of LepRb-null mice (*Donato et al., 2011*). In *Lep^ob* mice, GnRH fiber density is higher, but it decreases following leptin injection (*Bellefontaine et al., 2014*). A large density of synaptic contacts exists in the GnRH terminals in the vicinity of the ME, and specific inhibition of these terminals blocks LH pulses (*Wang et al., 2020*). We did not observe alterations in gonadotropin expression in the pituitary gland, but we found a reduction in *Tac2* levels, coding for Neurokinin B (NKB), in the hypothalamus of *Lepr^ΔVglut2* females. These findings indicate that blockade of glutamatergic neurotransmission in LepRb neurons disrupts key components of the GnRH pulse generator (*Han et al., 2023*; *Ruka et al., 2013*; *Uenoyama and Tsukamura, 2023*), leading ultimately to reproductive deficits. Indeed, previous research showed that activation of the Neurokinin 3 receptor (NK3R) by an NKB agonist-induced GnRH release consistently from the ME even in kisspeptin knockout animals, but not the preoptic region where GnRH cell bodies are located (*Gaskins et al., 2013*), suggesting that *Tac2* alone could be a direct mediator of the glutamatergic effect on GnRH release.

LepRb PMv neurons also project to the Arc and AVPV, where they contact Kiss1 neurons (*Donato et al., 2011*; *Ross et al., 2018*). However, the interaction and dynamics between the PMv and the kisspeptin populations remain unclear. In rats, the proestrus increase in *Kiss1* mRNA and Fos-ir in AVPV is blunted following bilateral lesions of the PMv. In the Arc, *Kiss1* mRNA is not altered during a leptin-induced LH release or in PMv lesioned animals (*Donato et al., 2009*; *Donato et al., 2013*). In mice, Arc Kiss1 neurons are downstream of PACAP PMv neurons in reproductive function (*Ross et al., 2018*). Thus, complementary to fast-acting glutamatergic neurotransmission, neuropeptides released from PMv cells contribute to the long-term regulation of reproduction by leptin and its integration with other modulatory signals, for example, the neuropeptide PACAP (*Evans et al., 2023*; *Ross et al., 2018*).

LepRb PMv neurons also express nitric oxide synthase (nNOS, *Nos1* gene) required for nitric oxide production (*Donato et al., 2011*; *Donato et al., 2010*; *Leshan et al., 2009*). Loss-of-function mutation in *NOS1/Nos1* gene in humans and mice cause infertility and blockade of nitric oxide neurotransmission disrupts leptin actin in reproductive axis (*Bellefontaine et al., 2014*; *Chachlaki et al., 2022*; *Yu et al., 1997*). Of note, glutamatergic neurotransmission is associated with activation of nitric oxide production via actions on glutamate N-methyl-D-aspartate (NMDA) receptor (*Chachlaki et al., 2017*; *d'Anglemont de Tassigny et al., 2007*; *Garthwaite et al., 1988*). Whether the reproductive deficits caused by disruption of glutamatergic neurotransmission in LepRb neurons are upstream of nitric oxide needs further investigation.

## Materials and methods

### Animals

The Lepr-null (*Lepr^loxTB*, kindly provided by Dr. Elmquist, UTSW Medical Center, Dallas, TX, available in JAX, stock 018989) (*Berglund et al., 2012*), the *Lepr^Cre* knock-in (*Lepr*-Cre, JAX; stock 032457) (*Leshan et al., 2006*), *Kiss1^hrGFP* (*Kiss1*-hrGFP; JAX, stock 023425) (*Cravo et al., 2013*), Vglut2^flox (*Slc17a6* or Vglut2 gene, JAX; Stock 012898) (*Tong et al., 2007*), and C57BL/6 (JAX; Stock 000664) mice were used for experiments. They were held in a 12:12 light:dark cycle (lights on at 6 AM), temperature-controlled (21–23°C) environment with ad libitum access to water and food. Mice were fed a low-phytoestrogen diet (Envigo 2016 diet) and a higher protein and fat phytoestrogen reduced diet (Envigo 2019 Teklad diet) when breeding. A phytoestrogen-reduced diet was used to avoid the exogenous effect of estrogen on reproductive physiology. Animals were bred and housed in our colony under the guidelines of the University of Michigan IACUC (PRO00010420). Procedures and experiments were carried out in accordance with the guidelines established by the National Institutes of Health 'Guide for the Care and Use of Laboratory Animals' and approved by the University of Michigan Committee on Use and Care of Animals.

### Stereotaxic injections

Unilateral stereotaxic injections of pAAV8.CMV.HI.eGFP-Cre.WPRE.SV40 (AAV-Cre) and pAAV2/7.CMV.PI.EGFP.WPRE.bGH (AAV-GFP) (Vector Core, University of Pennsylvania) in *Lepr^loxTB*, *Lepr^loxTB*;Vglut2^flox and wild-type females, and unilateral stereotaxic injections of pAAV8-hSyn-DIO-hM3D(Gq)-mCherry (AAV-hM3Dq, Addgene plasmid # 44361) (*Krashes et al., 2011*) from Bryan Roth, and of pAAV8-hSyn-DIO-mCherry (AAV-mCherry, Addgene plasmid # 50459) in *Lepr^Cre* females were

performed using a picospritzer attached to a glass pipette under isoflurane anesthesia in a stereotaxic apparatus (Kopf) (*Sáenz de Miera et al., 2023*). The stereotaxic coordinates for injection in PMv, measured from the rostral rhinal vein were [anteroposterior: −5.4 mm], the exposed superior sagittal sinus [mediolateral: ±0.5 mm], and from the top of the dura-mater [dorsoventral −5.4 mm]. Mice were followed twice daily for the first 3 days after surgery and daily for 7 more days for appropriate recovery. Experiments were performed 1 month after surgery.

## Activation of PMv LepRb neurons and its effect on LH release

Adult *Lepr^Cre* females (8–10 weeks old) received unilateral stereotaxic injections of ~50 nl of AAV-hM3Dq in the PMv. One month later mice were transferred to the Mouse Metabolic Phenotyping Center-Live (U2C – NIDDK) where they were cannulated in the aortic artery for serial blood sampling and the jugular vein for iv. injections (*Sáenz de Miera et al., 2023*). They were then connected to the Culex Automated Blood Sampling System 3 days after surgery. The following day, three blood samples were taken every 10 min to establish baseline hormonal levels, followed by iv. injection of either CNO ($n$ = 19) or clozapine ($n$ = 4) at (0.5 mg/kg in 50 μl, time 0). All injections were performed at the same time of the day, 1:00 PM. Blood samples were sequentially taken following the injection every 10 min for 1 hr. A negative control group received a stereotaxic injection of AAV-mCherry then a month later they were cannulated and the same protocol for blood collection was used. These animals received an injection of CNO on the day of the experiment ($n$ = 5). An additional group of *Lepr^Cre* females with no AAV injections were cannulated and received an iv. injection of clozapine ($n$ = 7) and were used as additional controls for blood collection. All animals were disconnected from the sampling system 2 hr after injection, a vaginal smear was collected to assess the phase of the estrous cycle and they were perfused under isoflurane anesthesia. Brains were collected and processed for histology. Blood samples were removed from the Culex machine immediately following the 2 hr mark and kept on ice. Samples were centrifuged at 14,000 × $g$ for 30 s at 4°C and plasma removed and stored at −80°C until hormone analysis.

## Assessment of body weight and reproductive phenotype

To inactivate glutamate in leptin responsive cells, *Lepr^Cre* mice were crossed with mice carrying loxP-modified *Slc17a6* alleles. Our experimental mice were homozygous for the *Lepr^Cre* allele (*Lepr^{Cre/Cre}*) and homozygous for the Vglut2-loxP allele (Vglut2^{fl/fl}). Our controls consisted of mice homozygous for the Cre allele (*Lepr^{Cre/Cre}*;Vglut2^{+/+}, named *Lepr^Cre*) or homozygous for the Vglut2-loxP allele (*Lepr^{+/+}*;Vglut2^{fl/fl}, named Vglut2^{flox}). Both experimental (*Lepr^{Cre/Cre}*;Vglut2^{fl/fl}, named *Lepr^{ΔVglut2}*) and control mice were derived from the same litters with parents homozygous for one of the genes and heterozygous for the other gene (*Lepr^{Cre/Cre}*;Vglut2^{fl/+} or *Lepr^{Cre/+}*;Vglut2^{fl/fl}). Mice were genotyped at weaning (21 days) and again at the end of the experiments.

Timing of puberty onset was monitored daily in experimental (*Lepr^{ΔVglut2}*) and control (*Lepr^Cre*) mice for external signs of puberty ($n$ = 7–13). In females, vaginal opening was monitored from postnatal day 19 (P19), followed by the timing for the occurrence of first estrus, a sign of puberty completion defined by the identification of keratinized cells for 2 consecutive days after 2 previous days with leukocytes in the vaginal lavage or by an estrus stage preceded by a proestrus stage (*Garcia-Galiano et al., 2017*). In males, the day of BPS was monitored from P25. Body weight for each animal was obtained on the day these events were observed. Animals were grown in normalized litters (5–7 pups/litter). Estrous cycles were assessed in young adult virgin mice. Females were housed with proven male breeders and monitored for fertility, latency to pregnancy, and number of pups per litter. Body weight of adult male and female mice was monitored weekly from 13 to 26 weeks of age. At the end of the experiment, total body weight, body fat, and lean mass were measured in conscious mice using a nuclear magnetic resonance-based analyzer (EchoMRI, 4in1-900).

## Kisspeptin injections

Adult *Lepr^Cre* and *Lepr^{ΔVglut2}* females (3–6 months) were transferred to the Mouse Metabolic Phenotyping Center-Live where they were cannulated in the aortic artery for serial blood sampling. Blood samples for baseline LH were taken every 7 min for 30 min before the injection. They then received an ip. injection of kisspeptin 65 μg/kg (kisspeptin-10, #048-56 Phoenix Pharmaceuticals) (*Wang et al., 2019*) and blood was then sampled every 7 min for 30 min.

## Endogenous reexpression of *Lepr* and concomitant deletion of *Slc17a6* in PMv neurons

The LepR null (*Lepr^loxTB^*) mouse model has a loxP-flanked transcription-blocking cassette (loxTB) inserted between exons 16 and 17 of the *Lepr* gene (**Berglund et al., 2012**; **Donato et al., 2011**). The *Lepr^loxTB^* mice lack the LepR intracellular domain and are obese, diabetic, and infertile. Cre-mediated excision of loxP sites generates LepR with signaling capacity. The *Lepr^loxTB^* mice were crossed with the Vglut2-floxed mice, which carry loxP sites flanking exon 2 of *Slc17a6*, to generate the *Lepr^loxTB^*;Vglut2^flox^ allowing for Cre-induced restoration of LepR and deletion of Vglut2 in PMv neurons. Mice were monitored for puberty and fertility as described above. Sixty days after introduction of the male breeders, females were anesthetized, perfused and brains were processed as described below.

Unilateral stereotaxic injections of AAV-Cre were used to restore LepR signaling in the PMv in homozygous *Lepr^loxTB^* 7- to 10-week-old females mice (previously described in **Mahany et al., 2018**) to restore LepR signaling and to selectively delete *Slc17a6* in the PMv of *Lepr^loxTB^*;Vglut2^flox^ homozygous females (*n* = 9). An additional cohort of *Lepr^loxTB^* was injected with AAV-Cre and included as the positive control (*n* = 5). WT mice were used as positive (fertile) controls (*n* = 13, six with AAV-GFP injection; seven without injections) (**Mahany et al., 2018**) and *Lepr^loxTB^* (*n* = 4) mice were used as negative (infertile) controls. Six to eight weeks after surgery, animals were ip. injected with leptin (3 mg/kg) 1.5 hr before perfusion. Brain, uterus, and ovaries were collected. Reproductive maturation was assessed by vaginal opening, uterus size and presence of ovarian corpus lutea. Brain tissue was processed for GFP immunohistochemistry and *Slc17a6* in situ hybridization (ISH).

### Tissue collection and histology

Mice were deeply anesthetized with isoflurane (Fluriso, VetOne) and intracardially perfused with 0.1 M phosphate-buffered saline (PBS) followed by 10% normal buffered formalin. Brains were dissected and kept in post-fixative (20% sucrose in fixative) for 4 hr and cryoprotected overnight in 20% sucrose in PBS. Brains were cut into four series of 30 µm coronal sections using a freezing microtome (Leica SM 2010R) and stored at −20°C in cryoprotectant until used. Brains of animals treated with leptin were postfixed for only 2 hr and were processed for pSTAT3-ir to validate LepR reexpression (**Münzberg et al., 2003**; **Williams et al., 2011**). Reproductive organs were fixed in 10% formalin and processed

**Table 2.** Antibodies.

| Antibody | Protein target | Host organism | Dilution used | Vendor and cat# | RRID |
|---|---|---|---|---|---|
| Anti-GFP (Chicken Antibodies, IgY Fraction) | Recombinant GFP null | Chicken | 1:10,000 | Aves Labs Cat# GFP-1010 | RRID:AB_2307313 |
| mCherry monoclonal (16D7) | mCherry tag | Rat | 1:5000 | Thermo Fisher Scientific Cat# M11217 | RRID:AB_2536611 |
| ABE457 | Anti-cFos | N-terminus cFos | Rabbit | 1:2000 | Millipore Cat# ABE457 | RRID:AB_2631318 |
| Pro-opiomelanocortin Precursor POMC (27-52) antibody | Aminoacids 27–52 of porcine POMC peptide null | Rabbit | 1:10,000 | Phoenix Pharmaceuticals Cat# H-029-30, | RRID:AB_2307442 |
| Anti LHRH (GnRH) | Synthetic LHRH | Rabbit | 1:1000 | Immunostar Cat# 20075 | RRID:AB_572248 |
| Phospho-Stat-3 (tyr705) | Phospho-Stat-3 (tyr705) | Rabbit | 1:1000 | Cell Signaling Technology Cat# 9131 | RRID:AB_331586 |
| Goat anti-Chicken IgY, Alexa Fluor 488 | Chicken, IgY (H+L) | Goat | 1:500 | Thermo Fisher Scientific Cat# A-11039 | RRID:AB_2534096 |
| Donkey anti-Rat IgG, AlexaFluor 594 | Rat IgG (H+L) | Donkey | 1:500 | Thermo Fisher Scientific Cat# A-21209 | RRID:AB_2535795 |
| Donkey anti-Rabbit IgG, AlexaFluor 488 | Rabbit IgG (H+L) | Donkey | 1:500 | Thermo Fisher Scientific Cat# A-21206 | RRID:AB_2535792 |
| Biotin-SP-conjugated AffiniPure Donkey Anti-Rabbit IgG | Rabbit IgG (H+L) | Donkey | 1:1000 | Jackson ImmunoResearch Labs Cat# 711-065-152 | RRID:AB_2340593 |

using paraffin-embedded protocol and Hematoxylin and Eosin (H&E) staining (*Donato et al., 2011*; *Mahany et al., 2018*).

## Immunofluorescence

Floating sections were rinsed with PBS and blocked with 3% normal donkey serum in PBS + % Triton X-100 for 1 hr at room temperature. Sections were incubated overnight at 4°C with primary antibodies in blocking buffer. Primary antibodies used are listed in *Table 2*. Sections were rinsed and incubated with secondary antibodies for 1 hr and rinsed in PBS. Secondary antibodies used are listed in *Table 2*. Sections were mounted on gelatin-precoated slides and coverslipped with Fluoromont-G (Electron Microscopy Sciences) mounting medium.

pSTAT3 immunoreactivity was performed to assess reactivation of functional leptin receptor. Brain sections were pretreated with 30% $H_2O_2$ to block endogenous peroxidase activity, 0.3% glycine, and 0.03% sodium dodecyl sulphate prior to primary antibody incubation. Tissue was blocked in 3% normal donkey serum + 0.25% Triton in PBS for 1 hr and incubated in primary rabbit anti-pSTAT3 (*Table 2*) for 48 hr at 4°C. Primary antibody was detected using a biotinylated secondary antibody (*Table 2*) by avidin–biotin peroxidase method (ABC Kit, Vector Laboratories) using diaminobenzidine (Sigma) as chromogen.

## In situ hybridization

Single ISH was performed to determine deletion of *Slc17a6* in PMv neurons ($n$ = 9 experimental vs. $n$ = 4 control). Briefly, a series of brain sections were mounted onto SuperFrost plus slides (Thermo Fisher Scientific), fixed in 10% NBF for 20 min and cleared with xylene for 15 min. Slides were boiled in sodium citrate buffer (pH 6.0) for 10 min. The *Slc17a6* DNA template was generated from mouse hypothalamic RNA by polymerase chain reaction(PCR) amplification. The following primers were used to amplify a 909 base-pair sequence in the gene encoding *Slc17a6* (exon 2 of the *Scl7a6* gene): forward (5′ CCGGGGAAAGAGGGGATAAAG 3′) and reverse (5′ GTAGACGGGCATGGATGTGA 3′). The antisense radio-labeled $^{35}$S- *Slc17a6* riboprobe was generated by in vitro transcription using a T7 RNA polymerase (Promega). $^{35}$S-labeled riboprobe was diluted in hybridization solution (50% forma-mide, 10 mM Tris–HCl pH 8.0, 5 mg tRNA, 10 mM dithiothreitol, 10% dextran sulfate, 0.3 M NaCl, 1 mM ethylenediaminetetraacetic acid (EDTA), and 1× Denhardt's solution), and brain slices were hybridized overnight at 57°C. Slides were then incubated in 0.002% RNase A followed by stringency washes in SSC (sodium chloride-sodium citrate buffer). Sections were dipped in NTB autoradiographic emulsion (Kodak) and stored in light-protected slide boxes at 4°C for 2 weeks. Signal was developed with developer and fixer (Carestream, Rochester, NY, USA), and the slides were cover-slipped with DPX (Electron Microscopy Sciences) mounting medium.

To confirm the deletion of *Slc17a6* in LepR neurons we did chromogenic ISH using BaseScope Duplex ($n$ = 4 *Lepr$^{Cre}$* vs. $n$ = 4 *Lepr$^{\Delta Vglut2}$*). Fresh frozen 16 μm coronal sections were collected on Superfrost Excell slides (Fisher) and stored at −80°C. Tissue sections were fixed in 10% NBF for 30 min and then dehydrated with a series of graded alcohols. The rest of the procedure was done with BaseScope Duplex Reagent Kit (ACD, #323800). Briefly, endogenous peroxidase was blocked with $H_2O_2$ for 10 min and tissue was digested with Protease IV for 10 min at room temperature. Sections were hybridized with BaseScope probes targeting *Lepr* (BA-Mm-Lepr-1zz-st, catalog # 895341) and a probe designed against exon 2 of *Slc17a6* (BA-Mm-Slc17a6-1zz-st-C2, # 1240891-C2) for 2 hr at 40°C. Following hybridization, amplification, and detection were done using the BaseScope Duplex Detection Reagents ACD, #323810, and probes were detected using Green (*Lepr*) and Fast Red (*Slc17a6*) and counterstained with 50% Gill's hematoxylin for 30 s.

## Quantitative PCR

Vglut2$^{flox}$ ($n$ = 7) and *Lepr$^{\Delta Vglut2}$* ($n$ = 5) females in diestrus were euthanized to collect blood for basal LH analysis and dissect tissues for quantitative PCR (qPCR). Fresh frozen pituitaries and mediobasal hypothalamus were homogenized in Qiazol (QIAGEN) and total RNA was extracted using chloroform and precipitated with isopropanol. Complementary DNA (cDNA) was synthetized using MultiScribe reverse transcriptase and random primers (Invitrogen) from 1.5 or 2 μg of RNA and diluted to 10 ng/μl. Gene expression was analyzed by qPCR in 20 ng of cDNA in triplicate 10 μl reactions using SYBR Green Taq MasterMix (Bio-Rad), following the manufacturer's instructions in a CFX-384 Real-Time

**Table 3.** Quantitative PCR primers.

| Gene | Primer sequence | NCBI accession No. | Vendor |
|---|---|---|---|
| Actb | Fwd 5'-GGCTGTATTCCCCTCCATCG-3'<br>Rev 5'-CCAGTTGGTAACAATGCCATGT-3' | NM_007393.5 | IDT |
| B2m | Fwd 5'-TCTCACTGACCGGCCTGTAT-3'<br>Rev 5'-GATCACATGTCTCGATCCCAGT-3' | NM_009735.3 | IDT |
| Cga | Fwd 5'-CCTCAGATCGACAATCACCTG-3'<br>Rev 5'-AGCATGACCAGAATGACAGC-3' | NM_009889.2 | IDT |
| Fshb | Fwd 5'-CTGGTGCTGGAGAGCAATCT-3'<br>Rev 5'- ACTTTCTGGGTATTGGGCCG-3' | NM_008045.3 | IDT |
| Kiss1 | Fwd 5'-GGCAAAAGTGAAGCCTGGAT-3'<br>Rev 5'-GATTCCTTTTCCCAGGCATT-3' | NM_178260.4 | MWG\|operon |
| Kiss1r | Fwd 5'-TGGCTGGTTCCCCTGTTTTT-3'<br>Rev 5'-GCAGCCAGGTTAGCGATGTA-3' | NM_053244.5 | IDT |
| Gnrhr | Fwd 5'-TCTTCATCATCCCCCTCCTC-3'<br>Rev 5'-GGAGTCCAGCAGACGACAAA-3' | NM_010323.2 | IDT |
| Lhb | Fwd 5'-CCAGTCTGCATCACCTTCAC-3'<br>Rev 5'- CAGCTGAGGGCTACAGGAAAG-3' | NM_008497.2 | IDT |
| Pdyn | Fwd 5'-CGTTGCTGTCAAGATCTGTTG-3'<br>Rev 5'-AGGCAGTCCGCCATAACATT-3' | NM_001286502.1 | IDT |
| Tac2 | Fwd 5'-TCTGTGTGGGATGTAAAGGAGGG-3'<br>Rev 5'-GACAGCCGCAAACAGCATGG-3' | NM_001199971.1 | IDT |
| Tac3r | Fwd 5'-TAAAAGTCATGCCAGGCCGT-3'<br>Rev 5'-AGGTGACACCCATGATGAGC-3' | NM_021382.6 | IDT |

PCR thermocycler (Bio-Rad). The primers used are specified in *Table 3*. mRNA expression in mutant vs. control samples was determined using relative gene copy numbers by the comparative threshold (Ct) method. The fold gene expression was calculated as $2^{-\Delta\Delta Ct}$ and shown as a percentage of relative gene expression in the control group.

## Hormone profile

For LH assay, sera were sent to the University of Virginia Ligand Assay and Core of the Center for Research in Reproduction (Charlottesville, Virginia, USA), where serum LH was measured by ultra-sensitive LH ELISA. This in-house method is based on *Steyn et al., 2013*. The capture monoclonal antibody (anti-bovine LH β subunit, 518B7) was provided by Janet Roser, University of California. The detection polyclonal antibody (rabbit LH antiserum, AFP240580Rb) was provided by the National Hormone and Peptide Program (NHPP). HRP-conjugated polyclonal antibody (goat anti-rabbit) was purchased from DakoCytomation (Glostrup, Denmark; D048701-2). Mouse LH reference prep (AFP5306A; NHPP) was used as the assay standard. The Limit of Quantitation (Functional Sensitivity) was defined as the lowest concentration that demonstrates accuracy within 20% of expected values and intra-assay coefficient of variation (%CV)<20% and was determined by serial dilutions of a defined sample pool. Intra-assay %CV is 2.2%. Inter-assay %CVs were 7.3% (Low QC, 0.13 ng/ml), 5.0% (Medium QC, 0.8 ng/ml), and 6.5% (High QC, 2.3 ng/ml). Functional sensitivity was 0.016 ng/ml. with an assay reportable range of 0.016–4.0 ng/ml and intra-assay CV of 6.24%.

## Image and data analysis

Photomicrographs were acquired using Axio Imager M2 (Carl Zeiss Microscopy). Confocal micros-copy images were acquired using NIS-Elements on a Nikon N-SIM+ A1R microscope with a resonance scanner. For cell quantification, ×40 images were used. Neuronal counting and coex-pression analysis were done using the 'Freehand selection' tool to outline the areas of interest and the 'Multi-point selection' and 'Region of Interest manager' tools in ImageJ (NIH) to compare the same neurons across different color channels. Sections containing the PMv were analyzed (Bregma ~−2.3/−2.70, Paxinos Brain atlas) and adjacent nuclei were also analyzed to determine the degree

of viral contamination. Quantification of GnRH fibers was performed in 2 consecutive ×20 images of the Arcuate nucleus (Bregma ~−2.06, Paxinos Brain atlas). For each image, a background value was measured and subtracted from the image. A 250 μm wide × 140 μm high rectangle was placed on the edge of the 3V wall to cover Arc GnRH-ir, carefully avoiding the saturated expression in the tuberoinfundibular sulcus region, and integrated density (mean pixel intensity × area) of the signal was quantified.

Data are expressed as mean ± standard error of the mean. Data were assessed for normality. When data did not fit a normal distribution, they were transformed and analyzed by a parametric test or analyzed using a non-parametric test. Unpaired two-tailed Student's $t$-test was used for comparison between two groups. For AUC of LH levels after CNO/clozapine, baseline LH level was brought to 0, subtracting the −10 min LH value from all the timepoints for each animal. Then, AUC analysis was performed and only the positive area was considered for analysis. One-sample $t$-test analysis was used comparing data to a theoretical mean of 0. LH levels after kisspeptin injection were analyzed using AUC. One-way analysis of variance (ANOVA) was used to compare three or more groups. Body weight data were analyzed using repeated-measures two-way ANOVA followed by Sidak's post hoc multiple comparison test. A p value less than 0.05 was considered significant. Excel software (Microsoft, Inc) was used to organize and calculate data. Statistical analyses and graphs were performed using GraphPad Prism v.7 (GraphPad software, Inc). Photoshop CS6 (Adobe, Inc) was used to integrate graphs and digital images into figures. Only brightness, contrast, and levels were modified to improve data visualization.

## Acknowledgements

The Michigan Mouse Metabolic Phenotyping Center-Live is supported by NIH Center Grant U2C DK135066 (Mi-MMPC), P30 DK020572 (MDRC), and P30 DK089503 (MNORC), and the Ligand Assay and Analysis Core – University of Virginia is supported by R24 grant HD102061. This work was supported by R01 HD069702 to CFE and R21 HD109485 to CFE and CSM, NIH R01 DK056731 to MGM, and URM Pilot Grant from P30 DK089503 (MNORC) to CSM.

## Additional information

### Funding

| Funder | Grant reference number | Author |
| --- | --- | --- |
| Eunice Kennedy Shriver National Institute of Child Health and Human Development | R01 HD069702 | Carol F Elias |
| Eunice Kennedy Shriver National Institute of Child Health and Human Development | R21 HD109485 | Carol F Elias<br>Cristina Sáenz de Miera |
| National Institute of Diabetes and Digestive and Kidney Diseases | R01 DK056731 | Martin G Myers |
| Michigan Nutrition and Obesity Research Center | URM Pilot Grant P30 DK089503 | Cristina Sáenz de Miera |

The funders had no role in study design, data collection, and interpretation, or the decision to submit the work for publication.

### Author contributions

Cristina Sáenz de Miera, Methodology, Writing – review and editing, Conceptualization, Data curation, Formal analysis, Validation, Investigation, Visualization, Writing - original draft, Project administration; Nicole Bellefontaine, Conceptualization, Formal analysis; Susan J Allen, Writing - original draft, Project administration; Martin G Myers, Investigation, Project administration; Carol F Elias,

Methodology, Writing – review and editing, Conceptualization, Supervision, Funding acquisition, Formal analysis, Investigation, Visualization, Writing - original draft, Project administration

**Author ORCIDs**
Cristina Sáenz de Miera ⓘD http://orcid.org/0000-0001-8047-035X
Carol F Elias ⓘD http://orcid.org/0000-0001-9878-9203

**Ethics**
Animals were bred and housed in our colony under the guidelines of the University of Michigan IACUC (PRO00010420). Procedures and experiments were carried out in accordance with the guidelines established by the National Institutes of Health 'Guide for the Care and Use of Laboratory Animals' and approved by the University of Michigan Committee on Use and Care of Animals.

Reviewer #1 (Public Review): https://doi.org/10.7554/eLife.93204.3.sa1
Reviewer #2 (Public Review): https://doi.org/10.7554/eLife.93204.3.sa2
Reviewer #3 (Public Review): https://doi.org/10.7554/eLife.93204.3.sa3
Author response https://doi.org/10.7554/eLife.93204.3.sa4

## Additional files

### Supplementary files
- MDAR checklist
- Source data 1. All data measured and analyzed for *Figures 1–3*.

### Data availability
All data generated or analyzed during this study are included in the manuscript. Source data files have been provided for Figures 1–7.

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
