## [Editor Report · eLife assessment]

This **important** study reports that glutamate signaling in LepRb PMv neurons is necessary for leptin-dependent fertility. The data supporting the conclusion is **solid**. This work will be of interest to researchers in the fields of both reproductive and metabolic biology.

---

## [Referee Report · Reviewer #1 (Public Review)]

Summary:

In previous work the Elias group has shown that leptin sensing PMv neurons make connections with the neuroendocrine reproductive axis and are involved in reproductive function/s. Sáenz de Miera et al. build on this body of work to investigate the sufficiency of leptin sensing PMv neurons to evoke the release of luteinizing hormone. The team further investigates how glutamate signaling from leptin-sensing neurons can influence pubertal timing in females, along with mature estrous cycles. Genetic ablation of Slc17a6 (Vglut2) from LepRb-expressing cells resulted in a delay of the first estrus cycle post pubertal transition, along with a significantly lengthened estrous cycle in mature females. However, this deficit did not lengthen the latency to birth of the first litter in experimental dams. Restoration of leptin signaling in LepRb PMv neurons that was previously shown to induce puberty and instate reproductive function in LepRb knock-out female mice (Mahany et al., 2018). Here, Sáenz de Miera et al. use a combined genetic and viral strategy to demonstrate that glutamate signaling in LepRb PMv neurons is required for sexual maturation in LepRb knock-out female mice.

Strengths:

Most of the experiments performed in this manuscript are well justified and rigorously tested. The genetic method to simultaneously remove glutamate signaling and restore the leptin receptor in LepRb PMv neurons was well executed and showed that glutamate signaling in LepRb PMv neurons is necessary for leptin-dependent fertility.

Weaknesses:

Analysis of experimentally induced luteinizing hormone release could be confounded by spontaneous pulses of luteinizing hormone that are independent of LepRb PMv neurons.

---

## [Referee Report · Reviewer #2 (Public Review)]

Summary:

This is a very well-written manuscript by Saenz de Meira and colleagues on a careful study reporting on the key role of glutamate transporter vGlut2 expression in the neurons of the ventral perimammillary nucleus (PMv) of the hypothalamus expressing the leptin receptor LepRb in energy homeostasis, puberty, and estrous cyclicity. The authors first show using cre-dependent chemogenetic viral tools that the selective activation of the PMv LepRb induces luteinizing hormone (LH) release. Then the authors demonstrate that the selective invalidation of vGlut2 in LepRb-expressing cells in the all body induces obesity and mild alteration of sexual maturation in both sexes and blunted estrous cyclicity in females. Finally, the authors knock out vGlut2 in PMv neurons in which they reintroduce LepRb expression in an otherwise LepRb-null background using an AAV Cre approach. This latter very elegant experiment shows that while the sole re-expression of LepRb in PMv neurons in LepRb-null mice was shown before to restore puberty onset, deleting vGlut2 in LepRb-expressing PMv neurons blunts this effect.

Strengths:

The authors employ state-of-the-art methods and their conclusions are robustly supported by the results.

Weaknesses:

None identified. Only minor comments have been formulated.

---

## [Referee Report · Reviewer #3 (Public Review)]

Summary:

The authors examined the effects of glutamate release from PMv LepR neurons in the regulation of puberty and reproduction in female mice.

Strengths:

Multiple genetic mouse models were utilized to either manipulate PMv LepR neuron activities or to delete glutamate vesicle transporters from LepR neurons. The authors have been quite rigorous in validating these models and exploring potential contaminations. Most of the data presented are solid and convincing and support the conclusion.

Comments on revised version:

The authors have addressed most of my comments.

---

## [Author Response]

The following is the authors’ response to the original reviews.

**Reviewer #1 (Recommendations For The Authors):**
I have one major concern regarding this draft of the manuscript:(1) In the manuscript (lines 130-31) it is stated that "About 55% (8/15) of mice with unilateral AAV-hM3Dq centered in the PMv showed an increase in LH release above 0.5ng/ml within 10-20 min following the CNO injection" However, data at time zero are not shown for 4 of the 8 "LH peak" animals. The missing data at time zero seems problematic for the analysis of the CNO-stimulated cohort. As mentioned in the manuscript, the area under the curve was calculated between the range of -10 to 20min post-injection. Because diestrus animals have spontaneous LH pulses, it is highly possible that an LH pulse is initiated in the10 minutes prior to drug delivery, as seen in the AAV-mCherry group in 1D, and similarly in 2C. Given the current form of analysis, it seems possible that a spontaneous LH pulse initiated anywhere up to 10 minutes prior to drug delivery could conceivably count as an experimentally induced "LH peak". Can you address this concern?

We understand the reviewer’s concern about the spontaneous LH pulses. This is the reason we have been very strict on our analysis and have taken multiple approaches to analyze these data. In our hM3Dq group 55% of the animals responded to CNO with an increase in LH, while 0 responded in the negative control group. But also, in the clozapine group, where no time 0 points were missing, 100% of the animals with hM3Dq showed an LH increase after the injection while only 28% (2/7) showed the increase in the negative control group. Rigorously, the DREADDs approach doubled the chances of LH increase. Note that the spontaneous LH peaks observed in negative controls or during baseline show a very sharp increase and decrease at the next time point, while the 4 “PMv hits” without time 0 and increase in LH in the CNO-hM3Dq group showed a sustained rise after the 10 min or prolonged high LH levels (above 1ng/ml) even 30 min after the injection. But, ultimately, the FOS levels in the PMv of CNO-hM3Dq group with increase in LH are significantly higher than in any other group and the number of FOS neurons are highly correlated to LH levels. Another important aspect that should not be dismissed is that in this experimental design, we used unilateral injection in animals that are in a fed state, therefore the leptin role in rising LH levels is probably dampened.

We have added a statement to clarify this issue.

The following are minor concerns:a) Figure 4 a-d, it is clear that Vglut2 is absent in the VMH, but it seems more relevant to show this expression pattern in the PMv.

We chose the VMH because it has a very dense collection of either LeprCre;VGlut2 or Vglut2 only cells and it illustrates very well the conditional Vglut2 deletion at small and high magnifications. In the PMv, however, the distribution of these cells is sparse. The reviewer is correct that for the current study, the PMv is more relevant and therefore, we have included images of the PMv showing a control and a LeprCre-Vglut2floxed animal in higher magnification.

b) Methods section, targeting PMv: please check the injection coordinate: "dura-mater [dorsoventral -0.54]"

Thank you for noticing this mistake, all coordinates for the injection have now been corrected (-5.4 mm, ±0.5 and -5.4mm)

**Reviewer #2 (Recommendations For The Authors):**
This is a very well-written manuscript by Saenz de Meira and colleagues on a careful study reporting on the key role of glutamate transporter vGlut2 expression in the neurons of the ventral perimammillary nucleus (PMv) of the hypothalamus expressing the leptin receptor LepRb in energy homeostasis, puberty, and estrous cyclicity. The authors first show using cre-dependent chemogenetic viral tools that the selective activation of the PMv LepRb induces luteinizing hormone (LH) release. Then the authors demonstrate that the selective invalidation of vGlut2 in LepRb-expressing cells in the all body induces obesity and mild alteration of sexual maturation in both sexes and blunted estrous cyclicity in females. Finally, the authors knock out vGlut2 in PMv neurons in which they reintroduce LepRb expression in an otherwise LepRb-null background using an AAV Cre approach. This latter very elegant experiment shows that while the sole re-expression of LepRb in PMv neurons in LepRb-null mice was shown before to restore puberty onset, deleting vGlut2 in LepRb-expressing PMv neurons blunts this effect.My specific comments are as follows. Please note that none of them require additional experiments and that they can be answered by amending the text.(1) Please provide information on the serotypes and promoters of the AAVs used in the study to enhance reproducibility.

Thank you, serotypes and promoters have been added for all AAVs.

(2) Please reformulate lines 220-221. Indeed, this reviewer does not agree with the fact that balanopreputial separation (BPS) is a sign of puberty completion. BPS is merely a sign of the advancement of sexual maturation, akin to vaginal opening in females. In certain mouse strains, BPS coincides with mini puberty rather than puberty. The definitive sign of puberty completion involves the presence of spermatozoa in the vas deferens (equivalent to the first ovulation/first estrus in females).

Thank you for this remark, this statement has now been modified.

(3) The authors convincingly show that the potential contamination of the arcuate nucleus of the hypothalamus (ARH) with the AAV injections targeted to the PMv should not account for the DREADD-mediated activation of LH release. However, do the authors believe that DREADD activation of LepRb-expressing PMv neurons, inducing cFOS expression in these neurons, could also activate ARH kisspeptin neurons (which do not express LepRb) via transsynaptic action? Alternatively, do they posit direct activation of GnRH cell bodies in the preoptic region or GnRH axon/dendrites in the ARH/median eminence region?

Thank you for this comment. We don’t have enough evidence from this DREADDs experiment to make a strong prediction on the downstream pathways. However, as discussed, from the DREADDs khrGFP females, we observed very few kisspeptin cells expressing cFOS, reducing the evidence for a PMv to ARH kisspeptin action in this case. With the evidence from our LepR-Cre;Vglut2flox animals that showed no alterations in kiss1 gene expression but a strong decrease in GnRH release, we hypothesize that this acute activation of LH is mediated by direct inputs from PMv to GnRH neurons, while acknowledging the possible existence of alternative pathways. These arguments have been added to the discussion.

(4) This reviewer finds it intriguing that glutamatergic signaling is required for LepRb re-expression in the PMv to restore fertility. Given that the authors and others have shown that PMv neurons heavily express NOS1, the activity of which is known to heavily rely on glutamatergic NMDAR activation, the authors may want to contextualize their results in light of the recent study showing that NOS1 is found to be a new causative gene in people with congenital hypogonadotropic hypogonadism.

Thank you for the advice, we have added a paragraph discussing the possible involvement of nNos from PMv neurons in the discussion.

(5) Does the absence of vGlut2 have any impact on the obesity phenotype in mice where LepRb is selectively re-expressed in the PMv?

We have followed the weight of these animals after the AAV injections. However, due to the difficulty of generating dual homozygous (LepRnull homozygous are infertile) and producing adequate stereotaxic injections with minimum contamination of adjacent nuclei, the groups could not be run all together and thus, we refrained from performing comparative analysis of energy balance. Analysis of body weight in LepRnull mice with reactivation of LepR in PMv neurons have been published before (Donato et al., 2011 using the Flp/Frt model and Mahany et al., 2018 using the Cre/loxP system). No difference in body weight was observed in both studies. Below is the progression of body weight in mice with reactivation of LepR and deletion of Vglut2 in PMv neurons. We added a comment on this regard.

**Author response image 1. sa4fig1:** 

**Reviewer #3 (Recommendations For The Authors):**
The authors examined the effects of glutamate release from PMv LepR neurons in the regulation of puberty and reproduction in female mice. Multiple genetic mouse models were utilized to either manipulate PMv LepR neuron activities, or to delete glutamate vesicle transporters from LepR neurons. The authors have been quite rigorous in validating these models and exploring potential contaminations. Most of the data presented are solid and convincing, and support the conclusion. This reviewer has the following suggestions for the authors to further improve this work and the manuscript.(1) The DREADD study had some issues. For example, "2 out of 7 control mice with no AAV showed an increase in LH...", indicating that LH increase may just happen randomly. More importantly, 45% of PMv-hit mice did not show LH response to CNO, making it hard to interpret the positive LH responses from the other 55% PMv-hit mice undergoing the same treatment. Overall, there are just too many variabilities in these DREADD data for anyone to come up with a clean and convincing conclusion. This reviewer suggests repeating these experiments or removing the DREADD data altogether. After all, the rest of the results are much more convincing and stand alone to support the role of glutamate release from these PMv LepR neurons.

We appreciate the reviewer’s concern. Indeed, LH shows spontaneous pulsatility which is one of the biggest challenges in our field. We have answered this concern for Reviewer 1 above and modified the text accordingly. We decided to keep the data in the publication because we believe that this is very important evidence supporting our observations since this is the only experiment that approaches the role of the PMv in a free-moving, ad libitum fed mouse model that is not deficient for leptin signaling or glutamatergic neurotransmission. Altogether this paper strongly supports a role for glutamate signaling on leptin’s action in reproductive function. Evidence for this role were dismissive or contentious until now.

(2) The mCherry signals in Figure 3 are of low quality and do not look like cell bodies.

We have now equally increased the contrast and brightness in all higher magnification images of mCherry neurons (Fig 3F, G, I and J) to improve their visibility. The lower magnification images are high quality images of areas with high density of mCherry positive neurons. Thick section (30µm) at low magnification compromises the focus at different Z-axis levels. We feel that images 3E and 3H are important to define the location of cells in the arcuate nucleus. Colocalization and mCherry expression are clear in high magnification images.

(3) The validation of Vglut2 deletion in LepR neurons (Fig. 4A-D) is very nice and convincing, but the images are from the VMH region. Why not show the PMv region?

As mentioned to Reviewer 1, we chose the VMH because it has a very dense collection of either LeprCre;VGlut2 or Vglut2 only cells and it illustrates very well the Vglut2 deletion at small and high magnifications. In the PMv, however, the distribution of these cells is sparce. The reviewer is correct that for the current study, the PMv is more relevant and therefore, we have included images of the PMv showing a control and a LeprCre-Vglut2floxed animal in higher magnification.

(4) Figures 4-5 used LepR-Cre as controls, while Figure 6 used Vglut2flox as controls. Why? Also, how did the authors set up the breedings to generate "littermates" in each of these studies?

We used the LepR-Cre as controls for our experiments since we need Cre homozygous for proper Cre expression and we had the LepR-Cre homozygous colony from the DREADDs experiment. Also, these mice had previously been thoroughly evaluated and no metabolic and/or reproductive disruption were noticed (please, see lines 213-214 of the original submission). However, our LepR-Cre colony had to be drastically reduced during COVID and suffered from unexpected Δ recombination leading to loss of Vglut2 homozygotes. To overcome these issues, we used VGlut2-floxed controls for the gene expression and GnRH immunoreactivity experiments. These mice had previously been used as controls for metabolic experiments with the LepCre-Vglut2fl genotype (Xu et al., 2013 Mol Metab), showing no deficiencies in the metabolic phenotype.

As described in the methods section (lines 464-466 of the original preprint), to inactivate glutamate in leptin responsive cells, LepRb-Cre mice were crossed with mice carrying loxP-modified Vglut2 alleles. Our experimental mice were homozygous for the LepRb-Cre allele (LepRb_cre/cre_) and homozygous for the Vglut2-loxP allele (Vglut2_fl/fl_). Our controls consisted of mice homozygous for the Cre allele (LepRb_cre/cre_;Vglut2_+/+*, named LepRb-Cre) or homozygous for the Vglut2-loxP allele (LepRb*+/+*;Vglut2_fl/fl*, named Vglut2_flox_). Both experimental (LepRb_cre/cre_;Vglut2_fl/fl_, named LepRbΔVglut2) and control mice were derived from the same litters with parents homozygous for one of the genes and heterozygous for the other gene (LepRb_cre/cre_;Vglut2_fl/+*or LepRb_cre/+*;Vglut2_fl/fl_). Mice were genotyped at weaning (21 days) and again at the end of the experiments.

(5) The labeling of Figures 5E-F is missing, making it hard to read.

We have confirmed that Figure 5E and F were mentioned in the figure legends and in the results text. To improve the analysis of the figure we have added the Y axis titles to Figure 5 C,D, E and F, previously only shown in Fig 5A and B.

(6) The last experiment was very nice confirming the role of glutamate release from PMv LepR neurons. However, the key phenotypes (puberty development, pregnancy) were not graphed and only stated in the text.

Thank you for your comment. Since the key result is that none the LeprLoxTb;Vglut2flox animals showed vaginal opening or pregnancy, we don’t feel the need to graph this. All the details of the reproductive and metabolic phenotyping of the Lepr-loxTB with re-expression of LepR in the PMV were described in Mahany et al., 2018.